# Foundational Automatic Evaluators: Scaling Multi-Task Generative Evaluator Training for Reasoning-Centric Domains

**Austin Xu,**\* **Xuan-Phi Nguyen,** **Yilun Zhou,**\* **Chien-Sheng Wu,** **Caiming Xiong,**\* **Shafiq Joty**

Salesforce AI Research
Correspondence: sjoty@salesforce.com

## ABSTRACT

Finetuning specialized generative evaluators has emerged as a popular paradigm to meet the increasing demand for scalable evaluation during both training and test-time. However, recent work has largely focused on applying new methodology, such as reinforcement learning (RL), to training evaluators, shying away from large-scale, data-driven development. In this work, we focus on data scaling, curating a set of 2.5M samples spanning five unique evaluation tasks (pairwise, step-level, reference-free and reference-based verification, and single rating) and multiple domains focused on reasoning evaluation. With our data, we train Foundational Automatic Reasoning Evaluators (FARE), a family of 8B and 20B (with 3.6B active) parameter evaluators, with a simple iterative rejection-sampling supervised finetuning (SFT) approach. FARE-8B challenges larger specialized RL-trained evaluators and FARE-20B sets the new standard for open-source evaluators, surpassing specialized 70B+ evaluators. Beyond static benchmarks, we evaluate FARE in real-world tasks: As inference-time rerankers, FARE-20B achieves near-oracle performance on MATH. As verifiers in RL training, FARE improves the downstream RL-trained model performance by up to 14.1% vs. string-matching verifiers. When initialized from FARE, a continually-finetuned FARE-Code outperforms gpt-oss-20B by 65% on evaluating test-case quality.

🤗 The FARE family of evaluators

Figure 1: Overview of our work. We curate 2.5M multi-task, multi-domain training samples (left) and use large-scale iterative rejection sampling SFT to train **FARE**, a family of automatic evaluators (top right). We evaluate FARE on static benchmarks and on various real-world downstream tasks.

---

\*Work done at Salesforce AI Research

## 1 INTRODUCTION

The past two years has seen the rapid adoption of large language models (LLMs) as automatic evaluators in response to demands for scalable evaluation of LLM outputs. LLM-based evaluators serve as judges for popular benchmarks (Dubois et al., 2024; Zheng et al., 2023), generative reward models for preference optimization (Yuan et al., 2024; Wu et al., 2024), and verifiers/critics in inference-time scaling settings (McAleese et al., 2024; Zhou et al., 2025b). The widespread integration of evaluators into nearly every phase of the LLM development cycle (Wu, 2025) demands evaluators that can handle multiple *evaluation tasks* while operating effectively across diverse *domains*.

Different settings require different evaluation abilities: Alignment needs evaluators capable of comparing different responses, i.e., pairwise evaluation, whereas monitoring model outputs requires finding minute mistakes, i.e., step-level evaluation. More recently, generative evaluators are tasked with providing reward signals during reinforcement learning (RL) training (Team Kimi et al., 2025; Jiang et al., 2025b), and are expected to grow in importance as RL moves towards unverifiable domains (Gunjal et al., 2025; Jayalath et al., 2025) in complex reasoning settings (Ke et al., 2025a; Ferrag et al., 2025). As evaluators take central roles in training and evaluating the next generation of models, they must be flexible enough to evaluate as the setting demands.

Compounding the challenges of multi-task evaluation is the expanding number of *domains* requiring evaluation: RL-training has quickly moved from math reasoning (e.g., Yu et al. (2025b)) to general-purpose reasoning (Ma et al., 2025) (e.g., history or economics). Agentic settings introduce additional wrinkles: With autonomously acting single agents (OpenAI, 2025; Nguyen et al., 2025; Wei et al., 2025) and complex multi-agent workflows (Liang et al., 2025; Alzubi et al., 2025) now being set free to browse the web and act on behalf of users with minimal oversight, evaluators must assess not only agent reasoning, but also proposed tool-use. These systems, sometimes built with intricate *model-generated* (Hu et al., 2024b; Zhang et al., 2024a; Ke et al., 2025b) interdependencies, are bottlenecked, in part, by subpar evaluation (Cemri et al., 2025).

Unfortunately, recent work in the open-source automatic evaluation community has failed to meet these twin demands of *multi-task, multi-domain* evaluators, opting instead in training task-specialized evaluators at relatively small data scales. We break this trend by *scaling up data*, curating 2.5M multi-task, multi-domain training samples that emphasize reasoning settings. As shown in Fig. 1, our data mix covers 5 distinct tasks and various domains like math, code, tool-use evaluation, and natural language reasoning. With our data, we train, **Foundational Automatic Reasoning Evaluators (FARE)**, two best-in-class evaluators. As shown in Fig. 1, our contributions are

- **Multi-task, multi-domain dataset:** We curate a large-scale, multi-task training set with an emphasis on reasoning-centric settings. We supplement existing human-and model-annotated data with synthetic data created from challenging new seed datasets.
- **Scalable learning via iterative rejection sampling:** We show that iterative rejection sampling supervised finetuning (RS-SFT) is a stable approach for training evaluators at scale. The semi-online nature of RS-SFT avoids problematic teacher model distribution shifts while bringing computationally stable and efficient model updates. Through ablations, we quantify the impact of training pipeline features like quantity of direct judgment data and the use of a continuous curriculum.
- **The FARE family of evaluators:** We train FARE-8B and FARE-20B and rigorously assess them with 7 challenging benchmarks and 3 practical downstream settings: test-time response reranking, RL-training verification, and domain-specific continual finetuning.

Our trained models are both well-rounded and high-performing. Out of the box, FARE improve generator performance at test-time, achieving near oracle reranking performance on MATH, and provide clear rewards during general-domain RL training, boosting downstream performance by 14.1% over typical string-matching verifiers. With minimal continual training, FARE can be adapted to specific domains like code, beating gpt-oss-20B by 65% in code test-case quality evaluation.

## 2 BACKGROUND AND RELATED WORK

An automatic evaluator (AE) $\pi_\theta : \mathcal{X} \to \mathcal{Y}$ maps input $x = (p, q, \mathcal{R}) \in \mathcal{X}$ to output $y = (c, j) \in \mathcal{Y}$. Input $x$ consists of $p$, the *evaluation protocol* that specifies both the *evaluation task* (e.g., pairwise comparison, verification) and evaluation rubric, $q$, the original question, and $\mathcal{R}$, set of model re-

sponses to be evaluated. The output $y$ consists of a natural language critique $c$ and final judgment $j$. The AE may also be prompted to omit the critique $c$ and directly output the judgment $j$, which we denote as $y = (\emptyset, j)$. The specific *evaluation protocol* $p$ determines the elements of set $\mathcal{R}$ and the exact form of judgment $j$. For example, in pairwise evaluation, $\mathcal{R}$ consists of two responses $\{r_1, r_2\}$ and the judgment is a choice between the two ("A" or "B"), whereas in single-rating, $\mathcal{R}$ consists of a single response $r$ and the judgment is an integer on a 1-5 scale. In this work, we focus on training automatic evaluators capable of the 5 evaluation tasks shown in Fig. 1:

- **Pairwise comparisons**: Given response set $\mathcal{R} = \{r_1, r_2\}$, the AE selects the better of $r_1$ and $r_2$.
- **Step-level evaluation**: Given response set $\mathcal{R} = \{r_{[steps]}\}$, where $r_{[steps]}$ is a single model response broken down into steps, the AE identifies step-level errors.
- **Reference-based verification**: Given response set $\mathcal{R} = \{r_{cand}, r_{ref}\}$, where $r_{cand}$ is the candidate and $r_{ref}$ is the reference, the AE determines if $r_{cand}$ is correct based on $r_{ref}$.
- **Reference-free verification**: Given response set $\mathcal{R} = \{r\}$, the AE determines if $r$ is correct.
- **Single rating**: Given response set $\mathcal{R} = \{r\}$, the AE assigns an integer score to $r$.

**Past work in generative automatic evaluators.** Capable LLMs, like GPT-4, were originally prompted as scalable evaluators (Wang et al., 2023a; Liu et al., 2023b; Fu et al., 2024; Chiang & Lee, 2023). Subsequent analysis revealed pitfalls of prompted approaches, like biases with respect to position (Wang et al., 2023b; Li et al., 2023), length (Zeng et al., 2023; Park et al., 2024), or self-preference (Panickssery et al., 2024). Finetuning specialized evaluators emerged as a result, with early approaches using teacher model outputs to do supervised finetuning (SFT) (Kim et al., 2023; 2024b; Li et al., 2023; Park et al., 2024; Shiwen et al., 2024) or direct preference optimization (DPO) (Hu et al., 2024c; Ye et al., 2024), often focusing only on one or two evaluation tasks. More recent methods moved to reasoning models as teachers (Khalifa et al., 2025).

Vu et al. (2024); Wang et al. (2024a); Cao et al. (2024); Alexandru et al. (2025) train *foundational evaluators* at larger data scales with multi-protocol capabilities via *offline* training methods like SFT or DPO. Such approaches take inspiration from general-purpose, large-scale multi-task learning (Sanh et al., 2021; Raffel et al., 2020; Wei et al., 2021), which showed broad generalization capabilities emerge with the scaling of training data. Foundational evaluators likewise were empirically shown to generalize to unseen evaluation tasks, prompts, and criteria while being more robust to common biases (Vu et al., 2024; Wang et al., 2024a).

Recent work has focused on *methodological* advances, either using inference-time scaling (Liu et al., 2025d; Chan et al., 2025; Zhao et al., 2025) or *online* training like reinforcement learning from verifiable rewards (RLVR) (Chen et al., 2025a;b; Whitehouse et al., 2025; Xu et al., 2025b; Xiong et al., 2025b) to improve evaluator performance. Because RLVR is computationally demanding with relatively brittle training pipelines (Guo et al., 2025; Yang et al., 2025), recent evaluators are typically trained on a small amount of data for a single task. Our work bridges early work in training foundational evaluators with more recent methodological advancements, demonstrating that a simple semi-online training approach enables stable multi-task training at scale.

**Desiderata for a new generation of evaluators.** Here, we outline our design philosophy for FARE. Beyond accuracy and robustness, we seek *efficiency*, as many evaluation settings like inference-time reranking or RL rollout verification demand low latency. In contrast to recent long chain-of-thought (CoT) evaluators (Chen et al., 2025b; Khalifa et al., 2025), we select base models with either no or very compact "thinking" CoTs. We also explicitly *avoid having the evaluator generate reference answers*. Past work has used evaluators to generate references during evaluation (Zheng et al., 2023; Li et al., 2024) or training rollout (Chen et al., 2025b). Not only does this risk severely degrading performance when the reference is wrong (Krumdick et al., 2025), it also converts evaluation into generation, turning a relatively easier task into a harder one (Zhou et al., 2025a).

## 3 FARE: DATA AND TRAINING RECIPE

### 3.1 DATA CURATION

We use two data approaches for curating our final training mix: Using **Existing** high quality training datasets created for evaluator and preference finetuning and generating **Synthetic** datasets through programmatic error injection and a generate-then-grade strategy.

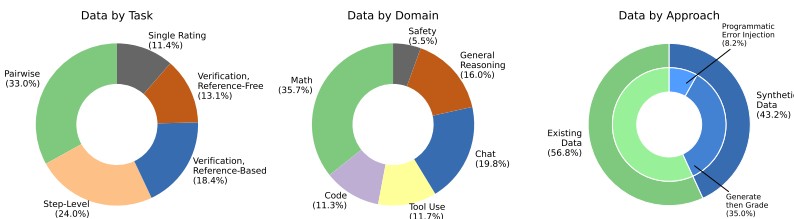

Figure 2: Breakdown of our curated training dataset of 2.5M samples by task (left), domain (center), and curation approach (right). Domain breakdown excludes step-level data, which is entirely math.

**Existing** data consists of training samples from proven sources that have produced effective evaluators (Vu et al., 2024; Cao et al., 2024). These datasets consist of high quality annotations from humans and frontier LLMs, and cover evaluation tasks like step-level, single-rating, and pairwise evaluation and domains like chat quality, code, and safety. Following Wang et al. (2024a), we largely focus our data collection on *modern* (2024 and beyond) datasets, as these datasets contain the most up-to-date model responses and fresh annotations. Beyond evaluator-specific training, we take advantage of existing preference fine-tuning datasets used for RLHF (Ouyang et al., 2022) and DPO training (Rafailov et al., 2023), converting these directly into pairwise evaluation samples. In domains with objectively correct answers, e.g., math, we also create verification training data with positive responses as correct/reference responses and negative responses as incorrect responses.

We hand-craft evaluation rubrics for each source dataset that follow annotation instructions given to human annotators or models, if existing. If such original instructions do not exist, we write custom evaluation rubrics for each source dataset based on the data composition and domain. App. E.2 provides an example rubric. **Existing** data lays a solid foundation, with 1.4M samples already dwarfing data scales found in recent work (e.g., 22K samples in Whitehouse et al. (2025) or 64K samples in Chen et al. (2025b)). However, upon inspection, we found three clear shortcomings: (1) Newly relevant tasks, like verification, were underrepresented. (2) Existing pairwise task data focused largely on chat-related topics and less-so on reasoning-relevant domains. (3) Questions and responses from newer, challenging datasets produced to meet the needs of reasoning-focused RL trainning were absent. To address these limitations, we supplement with synthetic data.

**Synthetic** data is generated from a diverse set of challenging seed datasets using two approaches:

- **Programmatic error injection.** We employ programmatic error injection when applicable, such as tool-use and function-calling data. For example, to create pairwise tool-use data, we inject errors (e.g., type error, extra argument, syntax errors) in correct function calls. This approach increases the amount of tool-use evaluation data, adding both pairwise and verification samples.
- **Generate-then-grade.** Here, we leverage a mix of recent and established training datasets comprised of question $q$ and verifiable ground-truth answer $a$. We sample up to 20 responses per $q$ from various generator models, then grade each responses based on $a$. After grouping responses by correctness, we create verification and pairwise. We use 12 unique generators from 6 model families, covering reasoning and non-reasoning models. This ensures that FARE are trained on a diverse array of model responses, enabling better generalization across distinct response distributions. Generate-then-grade enables us to incorporate problems from recent, challenging datasets covering frontier math and reasoning tasks. This enhances the quality of our pairwise and verification data with difficult-to-evaluate reasoning-focused samples.

Our final dataset comprises 2.5M training samples; an overview is shown in Fig. 2, with breakdowns by task, domain, and curation approach, and exact dataset sources are described in App. B.

## 3.2 MODEL TRAINING

**General training recipe.** We aim to train an automatic evaluator $\pi_\theta$, which we call the *policy model* parametrized by $\theta$. Our data curation process yields a training dataset of $N$ samples $\mathcal{D} = \{x_i, j_i^\star\}$, where $j_i^\star$ denotes the ground-truth judgment. Corresponding ground truth critiques $c^\star$ are not typically available in evaluator training data. As a result, past work has resorted to offline teacher-model based approaches or online RL-based approaches, as discussed in § 2.

These established approaches have their limitations: Teacher models introduce distribution shifts with respect to the policy model, a common problem with imitation learning (Ross et al., 2011), with choice of teacher model having a large impact on downstream performance (Guha et al., 2025). On the other hand, RL training is compute and time-intensive, making it difficult to scale to large data quantities, with past work (Liu et al., 2025c) only exploring small (1.5B parameter) models via ad-hoc interventions like reference policy resetting.

We borrow desirable qualities from both paradigms and use semi-online iterative rejection sampling SFT (RS-SFT) (Touvron et al., 2023; Dong et al., 2023), which was recently shown to be competitive with RLVR approaches (Xiong et al., 2025a). RS-SFT avoids sub-optimal distribution shift by finetuning on correct evaluation traces produced by the policy model while employing a computationally lightweight policy update step. This enables simple yet stable scaling to millions of training samples *without* sampling from a teacher model. An overview of our approach is shown in Fig. 1.

Concretely, we train $\pi_\theta$ as follows. We split our $N$ training samples into disjoint rollout batches $\mathcal{B}_t = \{x_{i,t}, j^\star_{i,t}\}$ of fixed size $N_{\text{rollout}}$ and initialize the initial policy $\pi_{\theta_0}$ to be an existing post-trained LLM (e.g., gpt-oss-20B). Then for step $t = 0, \dots, T - 1$, we perform the following:

- **Rollout from previous policy**: For inputs $x_{i,t}$ from $\mathcal{B}_t$, sample $K$ responses per input from policy $\pi_{\theta_t}$, denoted $\{\hat{y}^{(1)}_{i,t}, \dots, \hat{y}^{(K)}_{i,t}\}$.
- **Rejection sampling**: For each of the $K$ responses $\{\hat{y}^{(1)}_{i,t}, \dots, \hat{y}^{(K)}_{i,t}\}$, we determine correctness with ground-truth judgment $j^\star_{i,t}$. For inputs with correct responses, one randomly chosen response is kept. Any inputs without correct responses are discarded. We denote the collected set of inputs and corresponding correct responses as $\mathcal{D}_t$.
- **Policy update**: Use $\mathcal{D}_t$ to update the policy weights via SFT, initializing with $\theta_t$:

$$\theta_{t+1} = \arg\max_\theta \sum_{(x,y) \in \mathcal{D}_t} \log \pi_\theta(y|x) \tag{1}$$

Our approach draws inspiration from algorithms such as STaR (Zelikman et al., 2022) and RAFT (Dong et al., 2023), with some key differences. STaR notably re-initializes training from $\pi_{\theta_0}$ for each iteration $t$ and samples only one greedy response per input prompt, while RAFT relies on an external reward model to rank generated outputs. Because the automatic evaluation setting is inherently verifiable, i.e., the answer space of evaluators is closed vocabulary and discrete, like A/B for pairwise comparisons or yes/no for verification, we omit the need for a reward model to rank sampled responses.

Specific to evaluators, the Self-Taught Evaluator (STE) paradigm of Wang et al. (2024b) and follow-up EvalPlanner (Saha et al., 2025) are closely related to our approach. STE follows STaR with policy re-initialization and EvalPlanner uses multiple SFT and DPO training runs per iteration $t$. Further, these works use in-the-loop synthetic data generation, sampling responses to a small number ($< 25K$) of seed questions from a *fixed* generator. Pairwise samples are then created from correct/incorrect responses and used to train the model. This data generation approach cannot be adapted to create data for other tasks, like step-level evaluation, fundamentally limiting the task abilities of STE. A secondary concern is a lack of exposure to diverse response distributions; Evaluations in Wang et al. (2024a) show that scaling training data with a simpler training recipe leads to better generalization across benchmarks compared to STE.

**Batch composition.** For each rollout batch $\mathcal{B}_t$, we select unseen training samples from our curated training dataset, ensuring the task mixture is consistent with global task composition. For example, 33% of our overall training data are pairwise tasks (Fig. 2), so 33% of the input prompts in $\mathcal{B}_t$ are sampled from unseen pairwise samples. We then sample $K = 4$ responses per sample with a temperature of 0.9, and determine correctness based on final judgment.

**Inclusion of direct judgment data.** Past work (Wang et al., 2024a; Cao et al., 2024) has showed the importance of including *direct judgment* data samples to isolate judgment training signal. These are samples where the critique $c$ is omitted, and the input protocol $p$ is modified to prompt directly for a judgment. To precisely control the fraction of direct judgment data, we convert a fixed percent of $\mathcal{D}_t$ to direct judgment data by dropping generated critiques and modifying the input prompt accordingly. In App. D, we ablate the proportion of direct judgment data and show such data enables FARE to be prompted to exclude critiques for faster inference.

Table 1: Pairwise evaluation results, with **best** and second-best performance in each section marked. FARE achieve best-in-class performance, even outperforming frontier models in tool-use evaluation. † indicates that benchmark uses consistent accuracy (25% random baseline).

| | JudgeBench† | RJB† | PPE Correctness | RM-Bench | When2Call† |
|---|---|---|---|---|---|
| RISE-Judge-7B | 44.57 | 34.73 | 61.3 | 77.2 | 47.22 |
| EvalPlanner-8B | 30.20 | - | 52.8 | 68.1 | - |
| J1-8B | 42.00 | - | 59.2 | 73.4 | - |
| RM-R1-14B | 46.86 | 43.70 | **64.0** | 79.6 | 19.89 |
| CompassJudger-7B | 49.14 | 37.76 | 60.9 | **82.2** | 41.67 |
| Atla Selene 8B | 21.14 | 12.41 | 53.3 | 71.9 | 56.00 |
| CompassJudger-14B | 50.29 | 37.69 | 62.0 | 77.7 | 44.56 |
| **FARE-8B** | **55.71** | **51.05** | 63.8 | 79.2 | **80.33** |
| RISE-Judge-32B | 46.86 | 42.35 | 63.5 | 82.2 | 46.44 |
| CompassJudger-32B | 54.57 | 46.53 | 65.6 | 80.1 | 51.89 |
| RM-R1-32B | 54.29 | 46.39 | 65.9 | 81.5 | 23.89 |
| Self-Taught-70B | 48.3 | 38.64 | - | 73.6 | - |
| EvalPlanner-70B | 56.60 | - | 70.2 | 82.1 | - |
| J1-70B | 60.00 | - | 72.8 | 82.7 | - |
| **FARE-20B** | **64.29** | **57.05** | **74.4** | **90.5** | **76.67** |
| Qwen3-8B-ColdStart | 48.29 | 40.59 | 60.5 | 78.07 | 59.67 |
| Qwen3-8B | 52.27 | 43.56 | 64.8 | 79.9 | 64.78 |
| gpt-oss-20B | 59.43 | 50.51 | 71.7 | 89.9 | 61.33 |
| gpt-oss-120B | 70.29 | 58.26 | 77.8 | 92.0 | 70.00 |
| GPT-5-nano | 59.71 | 51.52 | 80.7 | 92.3 | 50.02 |
| GPT-5 | 84.86 | 79.57 | 87.0 | 93.8 | 75.78 |

Table 2: ProcessBench results, with **best** and second-best performance in each section marked. FARE-20B almost matches GPT-5 with the same prompt, achieving best-in-class performance, while FARE-8B beats comparably sized generative (Gen.) evaluators.

| | | GSM8K | MATH | OlympiadBench | OmniMATH | Overall |
|---|---|---|---|---|---|---|
| PRM | SkyworkPRM-1.5B | 59.0 | 48.0 | 19.3 | 19.2 | 36.4 |
| PRM | Math Shepherd-7B | 47.9 | 29.5 | 24.8 | 23.8 | 31.5 |
| PRM | SkyworkPRM-7B | 70.8 | 53.6 | 22.9 | 21.0 | 42.1 |
| PRM | ActPRM-7B | 82.7 | **82.0** | 72.0 | 67.3 | 76.0 |
| PRM | Qwen2.5-7B-PRM800K | 68.2 | 62.6 | 50.7 | 44.3 | 56.5 |
| PRM | Qwen2.5-Math-7B-PRM | 82.4 | 77.6 | 67.5 | 66.3 | 73.5 |
| PRM | Qwen2.5-Math-72B-PRM | **87.3** | 80.6 | **74.3** | **71.1** | **78.3** |
| Gen. | Qwen2.5-Math-7B | 26.8 | 25.7 | 14.2 | 12.7 | 19.9 |
| Gen. | RL Tango-7B | 53.1 | 48.2 | 37.8 | 36.3 | 43.9 |
| Gen. | StepWiser-1.5B | 46.9 | 43.4 | 26.3 | 28.4 | 36.3 |
| Gen. | StepWiser-7B | **72.4** | **68.3** | 54.4 | 52.4 | 61.9 |
| Gen. | **FARE-8B** | 68.5 | 67.7 | **59.9** | **58.1** | **63.5** |
| Gen. | Llama-3.3-70B | 82.9 | 59.4 | 46.7 | 43.0 | 58.0 |
| Gen. | Qwen2.5-Coder-32B | 68.9 | 60.1 | 48.9 | 46.3 | 56.1 |
| Gen. | QwQ-32B | 88.0 | 78.7 | 57.8 | 61.3 | 71.5 |
| Gen. | Qwen2.5-Math-72B | 65.8 | 52.1 | 32.5 | 31.7 | 45.5 |
| Gen. | GPT-4o | 79.2 | 63.6 | 51.4 | 53.5 | 61.9 |
| Gen. | **FARE-20B** | **89.8** | **87.8** | **80.0** | **79.9** | **84.4** |
| Gen. | Qwen3-8B-ColdStart | 37.0 | 41.0 | 36.3 | 38.9 | 38.3 |
| Gen. | Qwen3-8B | 63.2 | 64.0 | 51.5 | 48.2 | 56.7 |
| Gen. | gpt-oss-20B | 79.3 | 79.4 | 68.8 | 68.2 | 73.9 |
| Gen. | gpt-oss-120B | 89.6 | 87.6 | 80.8 | 76.0 | 83.5 |
| Gen. | GPT-5-nano | 83.8 | 87.0 | 80.6 | 77.1 | 82.1 |
| Gen. | GPT-5 | 91.4 | 89.5 | 80.6 | 76.9 | 84.6 |

**Per-batch continuous curriculum learning.** We additionally use a *continuous curriculum* in training: For each $(x, y) \in \mathcal{D}_t$, we compute the pass percentage from the $K = 4$ rollout generations for $x$, then sort the dataset in descending order of pass percentage. That is, samples where all 4 sampled outputs are correct are used to update the model first, and samples where only 1 of 4 sampled outputs are correct are used to update the model last. We find this has negligible impact on pairwise domains but large impacts in step-level evaluation, as we show in App. D.

**Base models.** We train two models starting from Qwen3-8B-Base (Yang et al., 2025) and gpt-oss-20B (Agarwal et al., 2025), denoted FARE-8B and FARE-20B. We find Qwen3-8B (post-trained) to be over-trained, and therefore cold-start Qwen3-8B-Base from SFT data from Qwen2.5-32B-Instruct, which we denote Qwen3-8B-ColdStart. See App. B.2 for additional details.

## 4 EXPERIMENTS

We evaluate FARE on both *core benchmarks*, static benchmarks for automatic evaluators, and in *downstream settings*, which simulate real applications of evaluators. We provide descriptions of benchmarks and baselines in App. C and additional ablations and analysis in App. D.

### 4.1 CORE BENCHMARKS

**Setup.** We evaluate along five diverse aspects: (*i*) *reasoning* with JudgeBench (Tan et al., 2024), ReasoningJudgeBench (RJB) (Xu et al., 2025b), and PPE Correctness (Frick et al., 2024), (*ii*) *bias and robustness* with RM-Bench (Liu et al., 2024b), (*iii*) *tool-use* with When2Call (Ross et al., 2025), (*iv*) *step-level error identification* with ProcessBench (Zheng et al., 2024), and (*v*) *reference-based verification* with VerifyBench (Yan et al., 2025).

For RM-Bench and PPE Correctness pairwise benchmarks, we adopt default evaluation setups, running each benchmark once with a fixed random ordering of responses. For other pairwise benchmarks, we report *consistent-accuracy* (Tan et al., 2024), where each test sample is run twice, swapping the order of response A and response B. If the evaluator selects a different response between runs (i.e., is positionally biased), then the sample is marked incorrect; if the evaluator is consistent, the judgment is graded against the ground-truth. For ProcessBench and VerifyBench, we report F1-score[1] and accuracy, respectively. We compare FARE against other finetuned generative and prompted evaluators. We report official numbers from past benchmarks, reporting sources in App. C. If necessary, we run each baseline using its own prompt template. For ProcessBench, we additionally compare against non-generative process reward models (PRMs).

**Results.** Tables 1 to 3 present our results on pairwise, step-level, and reference-based verification benchmarks, respectively. Our prompts are provided in App. E.

Table 1 shows that across diverse pairwise benchmarks, FARE exhibit best-in-class performance, outperforming comparably sized baselines. FARE-8B is the strongest small judge, outperforming recently released RL-trained models like J1-8B and RM-R1-14B by 13.71 and 6.57 absolute points on JudgeBench, respectively. FARE-20B challenges strong judges at 20B parameters, outperforming dense 70B-sized judge models despite having 3.5x fewer total parameters and nearly 20x fewer active parameters. The strong performance of FARE across reasoning benchmarks, which span math to scientific domains to causal reasoning, show that our models excel at discerning between objectively correct and incorrect responses. Beyond reasoning settings, FARE are generally robust to subtle, stylistic biases (RM-Bench) while excelling in tool calling evaluation (When2Call), which is increasingly important as agentic workflows grow in popularity.

Table 3: Ref.-based verification, with **best** and second-best performance marked per section. FARE beat general-purpose verifiers in hard settings.

|  | VerifyBench | VerifyBench-Hard |
|---|---|---|
| Math-Verify | 45.90 | 32.50 |
| GPT-4o mini | 92.85 | 72.30 |
| Llama-3.1-8B | 73.05 | 43.20 |
| Qwen3-4B | 92.00 | 72.40 |
| Phi-4 | 89.35 | 56.60 |
| Yi-1.5-9B | 87.70 | 61.40 |
| **FARE-8B** | **93.20** | **78.40** |
| GPT-4o | 93.15 | 72.60 |
| Llama-4-Scout | 90.01 | 48.50 |
| Llama3.3-70B | 83.25 | 54.70 |
| Qwen2.5-72B | 92.35 | 62.40 |
| Qwen3-32B | **95.80** | 71.80 |
| **FARE-20B** | 94.95 | **85.10** |
| Qwen3-8B-ColdStart | 92.45 | 72.60 |
| Qwen3-8B | 94.00 | 70.90 |
| gpt-oss-20B | 91.95 | 83.60 |
| gpt-oss-120B | 95.35 | 88.30 |
| GPT-5-nano | 94.65 | 84.00 |
| GPT-5 | 96.10 | 90.50 |

FARE also are extremely strong step-level evaluators, as shown by ProcessBench performance in Table 2. FARE-8B is the best small-sized generative critic model, outperforming the recently released StepWiser-7B (Xiong et al., 2025b), a RL-trained specialized step-level evaluator, by 1.6 points. FARE-20B outperforms the specialized Qwen2.5-Math-72B-PRM by 6.1 points, even matching GPT-5. Most notably, FARE excel on the two most challenging splits, OlympiadBench and Omni-MATH, with FARE-8B beating StepWiser-7B by 5.6 points and FARE-20B beating Qwen2.5-Math-

---

[1]ProcessBench defines their reported F1-score differently from the traditional F1-score; See this link.

Table 4: Scaling inference-time compute for FARE typically brings additional gains in performance: The performance gaps between FARE-8B/DeepSeek-GRM and FARE-20B/J1-70B widens.

| PPE | MMLU-Pro | MATH | GPQA | MBPP+ | IFEval | Overall |
|---|---|---|---|---|---|---|
| J1-8B w/ SC@32 | $65.6 \rightarrow 67.5$ | $70.0 \rightarrow 76.6$ | $53.2 \rightarrow 55.7$ | $53.1 \rightarrow 54.6$ | $54.0 \rightarrow 54.9$ | $59.2 \rightarrow 61.9$ |
| J1-70B w/ SC@32 | $79.0 \rightarrow 79.9$ | $86.0 \rightarrow 88.1$ | $65.9 \rightarrow 66.5$ | $66.0 \rightarrow \textbf{66.5}$ | $67.3 \rightarrow 67.2$ | $72.8 \rightarrow 73.6$ |
| DeepSeek-GRM-27B w/ SC@32 | $64.8 \rightarrow 65.5$ | $68.8 \rightarrow 69.4$ | $55.6 \rightarrow 56.0$ | $50.1 \rightarrow 49.9$ | $59.8 \rightarrow 61.0$ | $59.8 \rightarrow 60.4$ |
| DeepSeek-GRM-27B w/ MetaRM@32 | $64.8 \rightarrow 68.1$ | $68.8 \rightarrow 70.0$ | $55.6 \rightarrow 56.9$ | $50.1 \rightarrow 50.8$ | $59.8 \rightarrow 70.4$ | $59.8 \rightarrow 63.2$ |
| FARE-8B w/ SC@32 | $69.3 \rightarrow 70.8$ | $79.7 \rightarrow 80.4$ | $58.4 \rightarrow 58.4$ | $55.7 \rightarrow 54.9$ | $55.9 \rightarrow 56.7$ | $63.8 \rightarrow 64.2$ |
| FARE-20B w/ SC@32 | $80.3 \rightarrow \textbf{83.6}$ | $94.6 \rightarrow \textbf{97.3}$ | $68.5 \rightarrow \textbf{71.1}$ | $59.3 \rightarrow 57.3$ | $69.1 \rightarrow \textbf{71.1}$ | $74.4 \rightarrow \textbf{76.6}$ |

72B-PRM by 7.3 points on average. Overall, FARE are not only capable outcome-level evaluators, but are able to find subtle mistakes that manifest at the step-level.

FARE are also capable verifiers, as shown in Table 3, with both models outperforming all reported baselines on VerifyBench-Hard (Yan et al., 2025). In particular, FARE-20B excels beats the next best model, GPT-4o, by 12.5 absolute points. As we demonstrate in § 4.2, when used as verifiers during GRPO settings, FARE bring tangible benefits over typical string-matching verifiers.

**Scaling inference-time compute.** Recently, sampling parallel judgments and aggregating via majority vote, i.e., self-consistency (Wang et al., 2022), has been used to improve evaluator performance. In Table 4, we use self-consistency with 32 responses (SC@32) on PPE, comparing with J1 and DeepSeek-GRM (Liu et al., 2025d). DeepSeek-GRM also trains a MetaRM to perform judgment re-ranking at test-time. Across most splits, using SC@32 improves performance, with up to a 3.3 point improvement for FARE-20B on the MMLU-Pro split. Even without SC@32, FARE-8B beats DeepSeek-GRM-27B + MetaRM, with the gap widening with extra compute. Similarly, the gap between FARE-20B and J1-70B grows with extra compute. Interestingly, we see that across both our models, performance on MBPP+ slightly degrades, indicating that SC may not be the optimal way to use compute across all domains. Nonetheless, we observe gains in the aggregate.

## 4.2 DOWNSTREAM EVALUATION

**Setup.** We apply FARE on 3 downstream tasks: (*i*) Reward model for inference-time scaling, (*ii*) verifier for GRPO training, and (*iii*) initialization for continual finetuning for domain-specific evaluation. We provide detailed explanations of our downstream evaluation settings in App. C.2.

**Reward model for inference-time scaling.** We use the standardized setup in JETTS (Zhou et al., 2025b), which provides a set of 10 outputs from various generators and various benchmarks with corresponding correctness labels. Here, automatic evaluators rerank the responses, and performance is measured as the final performance of the evaluator-selected responses. We select the four most challenging benchmarks used in JETTS: MATH (Hendrycks et al., 2021), CHAMP (Mao et al., 2024), MBPP+ (Liu et al., 2023a), and BigCodeBench (Zhuo et al., 2024). We compare against strong judge models benchmarked previously on JETTS: SFR-Judge-8B,70B (Wang et al., 2024a), Skywork-Critic-8B,70B (Shiwen et al., 2024), Self-Taught-Evaluator-70B (Wang et al., 2024b), and

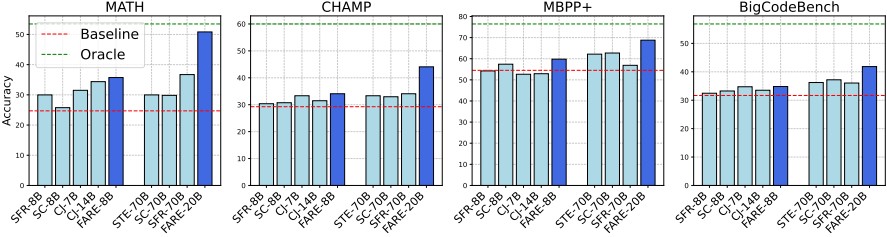

Figure 3: Best-of-10 performance for Llama-3.1-8B generator across 4 challenging benchmarks with baseline (red line) and oracle (green line) performance with FARE and SFR-Judge (SFR), Skywork Critic (SC), Compass-Judger (CJ), and Self-Taught Evaluator (STE) as baselines. FARE are the best small (≤14B) and large (≥ 20B) reranking models: FARE-20B achieves near oracle re-ranking performance on MATH, while FARE-8B matches 70B judges.

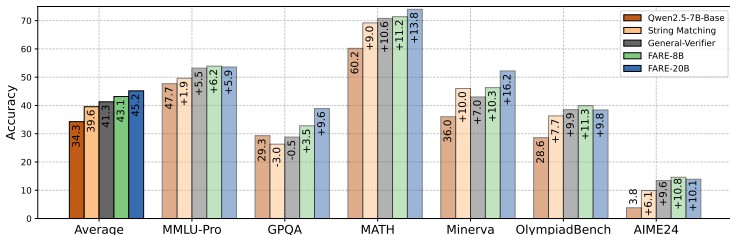

Figure 4: Performance of downstream GRPO-trained model with FARE as reference-based verifiers. Moving from string based output matching or weaker verifiers to FARE brings tangible performance gains across both natural-language based (e.g., GPQA) and math settings.

CompassJudger-7B,14B (Cao et al., 2024). We utilize the default pairwise reranking setup and prompt FARE to produce a judgment directly without explanation, i.e., $y = (\emptyset, j)$.

Fig. 3 shows best-of-10 performance with Llama-3.1-8B as generator with baseline greedy (red line) and oracle re-ranking performance (green line). FARE produce best-in-class reranking performance, with FARE-8B roughly matching the performance of larger (70B) judges in math settings and outperforming all similar sized judges in coding domains. FARE-20B excels in math domains, *approaching oracle-level reranking performance on MATH*, beating SFR-Judge-70B and Skywork-Critic-70B by 14 and 21 absolute points, respectively. Similarly, FARE-20B beats 70B+ judges by large margins in challenging coding domains. As we show in App. D.7, FARE improve the performance of other generators and FARE-20B improves significantly over gpt-oss-20B as a reranker.

**Verifer for GRPO training.** We train with WebInstruct-Verified (Ma et al., 2025), a multi-domain reasoning dataset, covering math, chemistry, etc. Verifier impact is measured via the downstream performance of the trained policy model on a fixed evaluation suite of MMLU-Pro (Wang et al., 2024c), GPQA-Diamond (Rein et al., 2024), MATH-500, Minerva-Math (Lewkowycz et al., 2022), OlympiadBench (He et al., 2024a), and AIME24 (Avg@32). We start from Qwen2.5-7B-Base (Yang et al., 2024) with the default reward setup as Ma et al. (2025) (see App. C.2), and we compare against training with the string-matching and trained verifier (General-Verifier) from Ma et al. (2025).

Training with FARE-20B as a verifier improves downstream performance from 34.3 to 45.2, a nearly 11 point absolute gain, with improvements coming uniformly across the six benchmarks, as shown in Fig. 4. Notably, with 77% fewer gradient updates[2], our model was able to slightly beat the performance of General-Reasoner-7B (Ma et al., 2025) on several benchmarks: 38.9 vs. 38.8 on GPQA-Diamond, 38.4 vs 37.9 on OlympiadBench, and 13.9 vs. 13.8 on AIME24. This shows that using FARE-20B can significantly improve RL training convergence. Further, using FARE-8B and FARE-20B bring 8.8% and 14.1% relative gains over typically used string matching verifiers and 4.4% and 9.4% over General-Verifier, which has been trained with in-training-distribution verification data. These gains appear for both natural language (MMLU-Pro, GPQA) and math domains, showing that FARE can verify complex outputs across multiple challenging domains.

**Initialization for domain-specific continual finetuning.** We continually finetune FARE-20B for code evaluation with one round of RS-SFT to produce FARE-20B-Code. We train with only 15K pairwise samples randomly chosen from Ace-Coder (Zeng et al., 2025). For evaluation, we use the recently released CodingJudgeBench (Jiang et al., 2025a), a pairwise benchmark covering code generation, code repair, and test-case quality evaluation tasks. Fig. 5, which reports consistent accuracy across the three splits from gpt-oss-20B, FARE-20B, FARE-20B-Code, and gpt-oss-120B for reference, shows that the first two splits are relatively easy, whereas test case evaluation is ex-

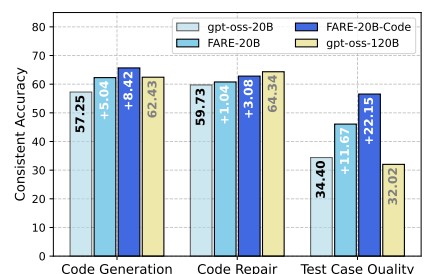

Figure 5: Continual training of FARE-20B for code evaluation with only 15K samples yields larger gains over gpt-oss-20B/120B.

---

[2]For the same group size, General-Reasoner-7B is trained for 700 steps with rollout batch size 768, whereas we train for 120 update steps with rollout batch size 1024.

tremely difficult: gpt-oss-120B achieves only 32%. On the last task, FARE-20B improves over gpt-oss-20B by 11.67 absolute points, highlighting the benefits of large-scale evaluation training. Specialized continual training on coding tasks brings an additional 10.48 absolute point improvement, with FARE-20B-Code outperforming even gpt-oss-120B on average. In all, FARE can be readily adapted for specific applications with a small amount of domain-specific data.

## 5 CONCLUSION

Using a curated multi-task, multi-domain training mix and RS-SFT, we train FARE, a family of high performing and well-rounded automatic evaluators. FARE-8B challenges larger specialized evaluators and FARE-20B sets a new standard for locally hosted evaluators. Our evaluations include 7 challenging benchmarks and 3 practical downstream settings where we show that FARE are (1) effective reward models at inference-time, (2) effective verifiers for GRPO training, and (3) strong initializations for continual, domain-specific finetuning.

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

APPENDIX

**Use of LLMs.** We minimally used LLMs during the writing process to (1) brainstorm revisions for short text snippets and (2) check for grammar, spelling, and writing mistakes.

**Reproducibility Statement.** We have released FARE checkpoints publicly and detailed training data sources in App. B.

**Ethics Statement.** The proliferation of LLM-based systems has raised concerns centered around model biases, faithfulness (e.g., hallucinations), and accuracy. As a result, automatic evaluation with LLMs has emerged as a popular paradigm for scalable evaluation of LLM outputs. Our work falls under this paradigm, training foundational automatic evaluators for complex reasoning settings.

Despite empirical successes, automatic evaluators are not a panacea for unreliable generators: Automatic evaluators themselves may not be free from bias or inaccuracies. Towards quantifying any bias in our evaluators, we evaluated FARE on RM-Bench, which aims to quantify how robust evaluators are to style and subtle mistakes in responses. However, we advise extensive bias testing and corrective finetuning before deploying automatic evaluators in the wild. When feasible, we advocate for having humans audit deployed evaluators for bias and systematic inaccuracies.

## A    EXTENDED RELATED WORK

Here, we cover recent advances in automatic evaluation that are non-generative. Scalar reward models (RMs) (Cobbe et al., 2021), which output a single scalar score for a given question $q$ and response $r$ were popularized originally within the context of reinforcement learning from human feedback (RLHF) (Ouyang et al., 2022; Stiennon et al., 2020; Ziegler et al., 2019). Here, pairwise human preferences were used to train a RM, which is then applied during RL-based alignment, e.g., with PPO (Schulman et al., 2017) or DPO (Hosseini et al., 2024). Past work has focused on dataset curation (Jiang et al., 2023; Liu et al., 2024a; 2025a; Dong et al., 2024) and experimenting with different loss formulations (Liu et al., 2024a; Lou et al., 2024).

The above class of reward models operate at the *outcome* level, with the entire response being assigned a single score. Step-level reward models, known as process reward models, attempt to provide denser feedback by assigning each "step" in a response a score. Training PRMs relies on step-level labels from humans (Lightman et al., 2023) or models (Duan et al., 2025) or using Monte Carlo simulation to estimate step-level quality (Wang et al., 2023c; Luo et al., 2024; Xiong et al., 2025b; Zhang et al., 2025). Both approaches have associated drawbacks: Step-level annotation is rarely scalable, while Monte Carlo simulation requires careful filtering to ensure high quality.

While RMs and PRMs remain popular paradigms, recent approaches have found that *generative* evaluators can better leverage test-time compute for stronger evaluation performance (Zhang et al., 2024b; Mahan et al., 2024; Liu et al., 2025d), inspiring recent advances in generative evaluators, as discussed in § 2.

## B    DATA AND TRAINING DETAILS

### B.1    TRAINING DATA

We enumerate our training data sources in Table 5 and present breakdowns by curation approach in Table 6. We took efforts to decontaminate our training sets with N-gram matching approaches, following Guha et al. (2025). For the **Synthetic** data approach, we use 12 unique generators, covering a mixture of weak and strong models: Ministral-8B (Team, 2024), Mistral-Small 24B (Team, 2025a), Gemma 3 12B (Team et al., 2025), Qwen2.5 7B, 32B (Yang et al., 2024), Qwen-QwQ (Team, 2025b), Qwen3-30B-A3B (Yang et al., 2025), Llama-3.1-8B, Llama-3.3-70B (Dubey et al., 2024), GPT-4o (Hurst et al., 2024), and GPT-4.1-nano, 4.1 (OpenAI, 2025) for seed datasets with verifiable answers. To increase diversity we randomly select a prompt template from a preset list for each question in the seed dataset and sample multiple responses at varying temperatures (e.g., 0.0, 0.3, 0.5, 0.7,...). For open-ended datasets, such as tool-use datasets, we enumerated common

Table 5: Data sources used to create our training set.

| Name | | Task | Domain | Curation Approach |
|---|---|---|---|---|
| ActPRM | Duan et al. (2025) | Step-Level | Math | **Existing** |
| Beavertails Preference | Ji et al. (2023) | Pairwise | Safety | **Existing** |
| Code Preference Pairs | Vezora/Code-Preference-Pairs | Pairwise | Code | **Existing** |
| DeepScaleR | Luo et al. (2025) | Pairwise, Verification | Math | **Synthetic** |
| Folio | Han et al. (2022) | Pairwise, Verification | NL Reasoning | **Synthetic** |
| FoVer | Kamoi et al. (2025) | Step-Level | Math | **Existing** |
| HelpSteer | Wang et al. (2023d) | Single rating | Chat | **Existing** |
| HelpSteer2 | Wang et al. (2024d) | Single rating | Chat | **Existing** |
| HelpSteer3 | Wang et al. (2025a;b) | Pairwise | Code, NL Reasoning | **Existing** |
| HH-RLHF Harmless | Bai et al. (2022); Ganguli et al. (2022) | Pairwise | Safety | **Existing** |
| LAMP | Chakrabarty et al. (2025) | Pairwise, Single rating | Chat | **Existing** |
| MATH | Hendrycks et al. (2021) | Pairwise, Verification | Math | **Synthetic** |
| MemGPT | MemGPT/MemGPT-DPO-Dataset | Pairwise, Verification | Tool-Use | **Existing** |
| OffsetBias | Park et al. (2024) | Pairwise | Chat | **Existing** |
| ReClor | Yu et al. (2020) | Pairwise, Verification | NL Reasoning | **Synthetic** |
| StepDPO | Lai et al. (2024) | Pairwise, Verification | Math | **Existing** |
| StrategyQA | Geva et al. (2021) | Pairwise, Verification | NL Reasoning | **Synthetic** |
| SWEGym | Pan et al. (2024) | Verification | Code | **Existing** |
| SWERank | Reddy et al. (2025) | Verification | Code | **Existing** |
| SynLogic | Liu et al. (2025b) | Pairwise, Verification | NL Reasoning | **Synthetic** |
| Tulu-V3-IF DPO data | Lambert et al. (2024) | Pairwise | Chat | **Existing** |
| WebDPO | WebDPO | Pairwise, Verification | NL Reasoning | **Existing** |
| When2Call Preference Pairs | Ross et al. (2025) | Pairwise | Tool-Use | **Existing** |
| XLam-60K | Liu et al. (2024c) | Pairwise, Verification | Tool-Use | **Synthetic** |

Table 6: Breakdown of training data by curation approach.

| Task | Pairwise | Step-level | Verification | Rating | | |
|---|---|---|---|---|---|---|
| Synthetic | 45.2% | - | 54.8% | - | | |
| Existing | 27.2% | 36.0% | 19.7% | 17.1% | | |
| Domain | General Reasoning | Math | Code | Tool-use | Chat | Safety |
| Synthetic | 28.3% | 52.1% | - | 19.6% | - | - |
| Existing | <1% | 50.4% | 14.7% | 1% | 25.7% | 7.2% |

errors found in tool calling outputs, such as invalid input types, missing arguments, malformed json format, etc., and then programmatically injected errors into ground-truth correct responses.

## B.2 TRAINING DETAILS

We train batch size 128 and a constant learning rate of 1e-6 and choose per-iteration rollout batch sizes of 50,000 and 250,000 for FARE-8B and FARE-20B, respectively. In the latter case, we make the practical trade-off of RS-SFT iterations for training speed, reducing the number of times we need to reset the model for rollouts, etc. We use a modified version of the OpenRLHF framework (Hu et al., 2024a) for training.

**Qwen3 cold-start SFT.** Public discussion from members of the Qwen organization indicate that post-trained versions of Qwen3 are difficult to continually finetune, and recommend starting from base models[3]. Therefore, we opt to cold-start SFT with one iteration of rejection sampling data collected from Qwen2.5-32B-Instruct. As we show in App. D.6, while this cold-start model does not match Qwen3-8B, FARE-8B outperforms the non-thinking Qwen3-8B on many static evaluation benchmarks.

We hypothesize that a relatively short, general-purpose alignment phase prior to evaluation-specific finetuning could further improve performance. While we did not attempt this, we believe this line of experimentation is of interest for future work.

---

[3]See, for example, this Twitter/X post: "...Instruct models after RL will pose difficulty for finetuning, but base models I don't think so..."

## C  Benchmark and Baseline Details

### C.1  Core benchmarks

For core benchmarks, we select a set of challenging and contemporary benchmarks for evaluating automatic evaluators:

- JudgeBench (Tan et al., 2024): A pairwise benchmark focused on evaluating LLM-as-judge models in reasoning settings, covering math, code, logical reasoning, and knowledge-based reasoning. Responses are generated using GPT-4o.

- ReasoningJudgeBench (Xu et al., 2025b): A pairwise benchmark that covers more diverse reasoning settings, such as multi-hop, causal, and domain-specific reasoning. Responses are generated using GPT-4o.

- PPE Correctness (Frick et al., 2024): A pairwise benchmark that covers reasoning and instruction following tasks with objectively correct answers, using seed datasets like MATH Hendrycks et al. (2021), GPQA (Rein et al., 2024), MBPP+ (Liu et al., 2023a), MMLU-Pro (Wang et al., 2024c), and IFEval (Zhou et al., 2023). Responses are generated using a variety of weaker models, e.g., Gemma-2-9B.

- RM-Bench (Liu et al., 2024b): A pairwise benchmark that evaluates how robust evaluators are to stylistic biases by evaluating on pairs of responses with subtle yet critical differences.

- When2Call (Ross et al., 2025): A pairwise benchmark that covers appropriate selection of tools (or refusals) in response to a user prompt. We use the LLM-as-judge test split, which comprises 300 unique prompts. Each prompt has four candidate answers (refusal, direct response, tool call, follow-up question), of which one response is correct. We form all pairs, yielding 900 total pairwise comparisons.

- ProcessBench (Zheng et al., 2024): A step-level benchmark that evaluates the ability to identify step-level errors in mathematical reasoning across easy (GSM8K and MATH) and hard (Omni-Math and OlympiadBench) questions.

- VerifyBench (Yan et al., 2025): A reference-based verification benchmark, comprised of Easy and Hard splits, that evaluates verifier ability to identify equivalent final answers.

For all core benchmarks, we utilize officially reported numbers when available. Otherwise, we run the corresponding baseline ourselves, using any prompt templates released with evaluators.

For pairwise benchmarks, we select our baselines from (1) existing multi-task foundational evaluators, (2) recently released RL-trained judge models, and (3) strong-performing specialized judges:

- RISE-Judge (Yu et al., 2025a): Pairwise judges trained with SFT then DPO to perform pairwise evaluation. Initialized from Qwen2.5 models.

- Self-Taught Evaluators (Wang et al., 2024b): A pairwise judge trained with iterative SFT with training data generated in the loop. Initialized from Llama-3.1-70B.

- EvalPlanner (Saha et al., 2025): Pairwise judges trained with iterative SFT and DPO on a small seed dataset, with an emphasis on learning how to plan for evaluation tasks. Initialized from Llama-3.3-70B.

- RM-R1 (Chen et al., 2025b): A family of pairwise judges trained with GRPO, initialized from DeepSeek-distilled Qwen models.

- J1 (Whitehouse et al., 2025): A pairwise and single-rating judge trained with GRPO. Initialized from Llama-3.1/3.3 models.

- CompassJudger (Cao et al., 2024): A family of foundational evaluators trained with large-scale SFT. Initialized from Qwen2.5 models.

- Atla Selene (Alexandru et al., 2025): A foundational evaluator trained with large-scale preference optimization. Initialized from Llama-3.1-8B.

We run gpt-oss variants with low reasoning, as (1) FARE-8B is trained initialized from gpt-oss-20B-low, and (2) evaluation often demands low-latency, making long CoT undesirable if they can be

avoided. For GPT-5, we use the default API settings (medium reasoning). For pairwise benchmarks, we use reported scores from Whitehouse et al. (2025); Xu et al. (2025b); Liu et al. (2025d).

For ProcessBench, we use officially reported numbers in Zheng et al. (2024), which includes Sky-workPRMs (He et al., 2024b), Math Shepherd PRM (Wang et al., 2023c), ActPRM (Duan et al., 2025), and Qwen-Math PRMs (Zhang et al., 2025). We additionally report results from generative baselines: RL Tango (Zha et al., 2025) and StepWiser (Xiong et al., 2025b). For VerifyBench, we use reported scores from the original paper (Yan et al., 2025).

## C.2 Downstream settings

**Reward model for inference-time scaling.** We compare our models against the following base-lines, representing best-in-class performers as reported in JETTS. We utilize reported numbers di-rectly except for CompassJudger, which was not included in the original JETTS evaluation. As such, we run CompassJudger ourselves.

- SFR-Judge-8B, 70B (Wang et al., 2024a): A family of multi-task evaluators. Among the highest performing small and large judges on JETTS. Initialized from Llama-3.1 models.
- Skywork-Critic-8B, 70B (Shiwen et al., 2024): Two pairwise-specfic evaluators that do not output explanations. Among the highest performing small and large judges on JETTS. Initialized from Llama-3.1 models.
- Self-Taught-Evaluator-70B (Wang et al., 2024b): A strong performing large judge model. Initialized from Llama-3.1 models.
- CompassJudger-7B, 14B (Cao et al., 2024): As described above.

**Verifier during GRPO training.** We adopt the settings of Ma et al. (2025), which train General-Reasoner, a family of reasoning LLMs of varying model sizes using the WebInstruct-Verified train-ing dataset. In particular, we train with standard GRPO, i.e., without dynamic sampling or clip higher modifications, initializing from Qwen2.5-7B-Base. We the same conditional reward setup as General-Reasoner:

- If the solution parsing fails, then reward is set to $-0.5$.
- If a solution successfully parsed and is deemed correct by the verifier, it is assigned a reward of 1 plus a length penalty of:

$$-0.05 \times \min\{10, |\texttt{len(ground\_truth)} - \texttt{len(model\_response)}|\}.$$

The training framework is based on the `verl` (Sheng et al., 2024). We use rollout batch size 1024, max response length of 4096, group size of 8, a temperature of 1.0, a KL coefficient of 0.001, and a learning rate of 5e-7.

**Initialization for domain-specific finetuning.** We randomly sample 15,000 pairwise samples from AceCoder (Zeng et al., 2025) and perform one round of rejection sampling. We adopt training setup of FARE-20B: a direct judgment ratio of 60% and continuous curriculum. We train for 3 epochs with batch size 256 and cosine decay learning rate peaking at 1e-5. We then evaluate on CodingJudgeBench, reporting consistency accuracy.

Note that CodingJudgeBench reports an unconventional pairwise metric, employing Z-score nor-malization between the two consistency runs. Their implementation is not publicly available, and their paper lacks concrete implementation details. As such, we resort to consistent accuracy, which is more commonly used in pairwise benchmarks, e.g., (Tan et al., 2024; Li et al., 2023; Xu et al., 2025a;b).

## D Ablations, analysis, and additional results

### D.1 Training recipe ablations.

In Table 7, we ablate three components of our training recipe and report the average on our five pairwise benchmarks and ProcessBench. First, we train multiple checkpoints using RS-SFT varying

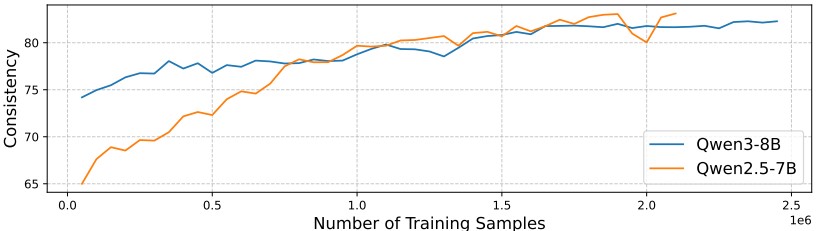

Figure 6: Pairwise positional robustness emerges as at large data scales, as shown with training FARE-8B and an earlier development run initialized with Qwen2.5-7B-Instruct.

the proportion of direct judgment data from 30% to 70%. Direct judgment data affects FARE-8B and FARE-20B differently: endpoints (30% or 60-70%) show the best performance for FARE-20B, whereas performance peaks at 40% for FARE-8B. As such, we choose 40% and 60% to train FARE-8B and FARE-20B.

We also ablate different strategies for training with direct judgment data specific to gpt-oss, which is trained to output an intermediate CoT before responding. We can either keep this CoT with direct judgment data or remove the intermediate CoT by going directly to the assistant turn. The former more closely mimics the training distribution of gpt-oss, but undermines the intended purpose of direct judgment data of isolating outcome correctness training signal, as discussed in § 3.1. The latter is out-of-training-distribution but more effectively isolates training signal. As seen in Table 7 (bottom right), removing the intermediate CoT leads to gains in both pairwise and step-level settings.

Finally, we measure the impact of the continuous curriculum as compared to a random data shuffling strategy with FARE-8B. As shown in Table 7 (bottom left), the continuous curriculum leads to minimal drops in pairwise performance but large gains in ProcessBench.

### D.2  HOW DOES DIRECT JUDGMENT PROMPTING AFFECT PERFORMANCE?

Many settings demand low latency, such as inference-time reranking or evaluating rollouts during RL training. Here, we study how performance varies when FARE are prompted to skip the critique $c$ and directly output a judgment $j$. For FARE-20B, this involves additionally skipping the intermediate CoT, directly outputting from the assistant turn, making out-of-distribution relative to gpt-oss-20B's original training setup. We see that performance *improves* for FARE-8B, but degrades for FARE-20B. Such results for FARE-8B are in-line with prior work, which finds that direct judgment-like inference leads to minimal drops in performance. For FARE-20B, we hypothesize that the post-training of gpt-oss-20B instills a strong prior in favor of generating intermediate CoT. Even training with direct judgment data, removing such CoT is detrimental. Nonetheless, performance does not universally degrade, with When2Call performance increasing by nearly 13 points.

### D.3  ROBUSTNESS TO PAIRWISE POSITIONAL BIAS EMERGES WITH DATA SCALE.

A known issue in pairwise evaluation is inconsistency (Wang et al., 2023b), a form of positional bias where the evaluator judgment changes based on the order of responses in the input prompt. During training, we observed that our judges become more consistent as a function of data scale; Fig. 6

Table 7: Ablation study varying proportion of direct judgment data, use of continuous curriculum (8B model), and ablating strategies for using direct judgment data for gpt-oss (20B model).

| % direct judgment data | Qwen3-8B-ColdStart Models | | | | gpt-oss-20B Models | | |
|---|---|---|---|---|---|---|---|
| | Pairwise | ProcessBench | Average | | Pairwise | ProcessBench | Average |
| 30 | 61.36 | 52.10 | 56.73 | | 72.51 | 83.64 | 78.08 |
| 40 | 61.14 | **58.03** | **59.59** | | 70.82 | 82.50 | 76.66 |
| 50 | 61.56 | 56.85 | 59.21 | | 71.97 | 83.00 | 77.49 |
| 60 | 62.31 | 54.73 | 58.52 | | **72.58** | 84.40 | **78.49** |
| 70 | **62.67** | 56.49 | 59.58 | | 71.59 | **84.72** | 78.16 |
| Curriculum | Pairwise | ProcessBench | Average | Keep CoT? | Pairwise | ProcessBench | Average |
| Yes | 61.14 | **58.03** | 59.59 | Yes | 67.80 | 81.64 | 74.72 |
| No | **61.49** | 53.43 | 57.46 | No | **69.81** | **82.55** | **76.18** |

Table 8: Performance with and without critiques (and CoT for FARE-20B) for pairwise benchmarks (left) and ProcessBench (right). Directly prompting for a verdict universally improves performance for FARE-8B, but degrades performance for FARE-20B.

| | JudgeBench | RJB | PPE Correctness | RM Bench | When2Call | Average | GSM8K | MATH | Olympiad Bench | Omni MATH | Average |
|---|---|---|---|---|---|---|---|---|---|---|---|
| FARE-8B | 55.71 | 51.05 | 63.8 | 79.2 | 80.33 | 66.0 | 68.5 | 67.7 | 59.9 | 58.1 | 63.5 |
| FARE-8B, no critique | 60.00 | 52.60 | 64.9 | 81.8 | 86.55 | 69.2 | 68.5 | 68.6 | 59.0 | 58.5 | 63.7 |
| FARE-20B | 64.29 | 57.05 | 74.4 | 90.5 | 76.67 | 72.6 | 89.8 | 87.8 | 80.0 | 79.9 | 84.4 |
| FARE-20B, no critique or CoT | 62.00 | 55.23 | 68.9 | 85.5 | 89.11 | 72.1 | 79.8 | 73.2 | 70.0 | 70.3 | 73.3 |

Table 9: Ablation comparing training task-specific evaluators against training a single multi-task evaluator. All models initialized from Qwen2.5-7B-Instruct.

| | JudgeBench | ReasoningJudgeBench | ProcessBench |
|---|---|---|---|
| Pairwise only | 53.71 | 48.78 | - |
| Step-level only | - | - | 55.32 |
| Multi-task | 58.00 | 51.11 | 55.81 |

shows the progression of pairwise consistency on the five pairwise benchmarks in § 4.1 over the course of training for FARE-8B and an earlier training run which was initialized from Qwen2.5-7B-Instruct. Both models steadily become positionally robust over the course of training, with the weaker Qwen2.5-7B-Instruct showing substantial gains. This reveals that *scaling evaluator training data can mitigate common judge biases*, complementing mitigation strategies that use data augmentation (Saha et al., 2025), label balancing (Cao et al., 2024; Wang et al., 2024a), and RL-based reward or algorithmic methods (Whitehouse et al., 2025; Xu et al., 2025b).

### D.4 MULTI-TASK EVALUATOR TRAINING OUTPERFORMS SINGLE-TASK TRAINING.

Here, we compare training evaluators with a multi-task data mix against training per-task evaluators. We initialize train from Qwen2.5-7B-Instruct, and train a pairwise only evaluator, a step-level only evaluator, and one with pairwise and step-level data. We train all models for an equivalent number of training input samples with RS-SFT on an earlier version of our final data mixture. We report results on JudgeBench, ReasoningJudgeBench, and ProcessBench in Table 9. We observe that multi-task training leads to significant gains over single-task evaluators, especially pairwise evaluators, with larger gains coming in pairwise evaluation settings that step-level evaluation. This result is intuitive: The skill of identifying errors at a granular (step) level improves pairwise evaluation by endowing the evaluator with the ability to catch more subtle mistakes in each response within the pair.

### D.5 SINGLE-RATING EVALUATION.

We additionally evaluate FARE on Single Rating tasks with BiGGen-Bench (Kim et al., 2024a) and FLASK (Ye et al., 2023), two chat-centric evaluation datasets with human annotated 1-5 ratings. We measure Pearson correlation with human annotations, and report results in Table 10. Single-rating is widely used evaluation task in reasoning settings, and thus constituting the smallest proportion of our training data, as shown in Fig. 2. Nonetheless, FARE are competitive with chat-focused judge models, with FARE-8B outperforming foundational judge models like SFR-Judge-8B and 12B and FARE-20B approaching the performance of SFR-Judge-70B. We use reported values from baseline papers, including LMUnit (Saad-Falcon et al., 2024), Atla Selene (Alexandru et al., 2025), and SFR-Judge (Wang et al., 2024a).

### D.6 COMPARISON AGAINST GENERAL-PURPOSE MODELS

Here, we compare FARE against general-purpose LLMs, selecting popular reasoning and non-reasoning models. Table 11 shows our results. Our cold-start SFT for Qwen3-8B produces the weakest Qwen3 variant, as expected. However, after undergoing iterative SFT, FARE-8B surpasses Qwen3-8B on multiple benchmarks, improving from 43.56 to 51.05 on ReasoningJudgeBench and 56.7 to 63.5 on ProcessBench. Likewise, we are able to improve gpt-oss-20B across the board, yielding substantial improvements in reasoning, tool-calling, and step-level evaluation. The resulting checkpoint approaches gpt-oss-120B on a number of benchmarks.

Table 10: Single rating performance, with **best** and second-best performance in each section marked. Despite being trained with a focus on reasoning settings, FARE perform competitively in single-rating evaluation in chat settings.

|  | FLASK | BiGGen Bench | Average |
|---|---|---|---|
| GPT-4o-mini | **0.630** | 0.600 | 0.615 |
| Glider-3.8B | 0.615 | 0.604 | 0.610 |
| FlowAI-Judge-3.8B | 0.400 | 0.460 | 0.430 |
| Prometheus-2-7B | 0.470 | 0.500 | 0.485 |
| Auto-J-13B | 0.350 | 0.300 | 0.325 |
| Themis-8B | 0.540 | 0.580 | 0.560 |
| SFR-Judge-8B | 0.520 | 0.590 | 0.555 |
| SFR-Judge-12B | 0.590 | 0.570 | 0.580 |
| Atla Selene 8B | 0.613 | 0.584 | 0.599 |
| LMUnit-8B | 0.600 | **0.645** | **0.623** |
| FARE-8B | 0.611 | 0.616 | 0.591 |
| GPT-4o | 0.690 | 0.650 | 0.670 |
| Prometheus-8x7B | 0.540 | 0.520 | 0.530 |
| SFR-Judge-70B | 0.660 | 0.650 | 0.655 |
| LMUnit-70B | **0.720** | **0.677** | **0.699** |
| FARE-20B | 0.649 | 0.616 | 0.633 |

Table 11: Comparison of FARE against their initial models and other popular general-purpose models. [†] indicates some results reported in Whitehouse et al. (2025) or Zheng et al. (2024).

|  | JudgeBench | ReasoningJudgeBench | PPE Correctness | RM-Bench | When2Call | Avg. consistency | ProcessBench |
|---|---|---|---|---|---|---|---|
| Qwen3-8B-ColdStart | 48.29 | 40.59 | 60.5 | 78.07 | 59.67 | 72.55 | 38.3 |
| Qwen3-8B-non-thinking | 52.27 | 43.56 | 64.8 | 79.9 | 64.78 | 74.04 | 56.7 |
| FARE 8B | 55.71 | 51.05 | 63.8 | 79.2 | 80.33 | 82.28 | 63.5 |
| gpt-oss-20B (low) | 59.43 | 50.51 | 71.7 | 89.9 | 61.33 | 77.83 | 73.9 |
| FARE 20B | 64.29 | 57.05 | 74.4 | 90.5 | 76.67 | 82.92 | 84.4 |
| gpt-oss-120B (low) | 70.29 | 58.26 | 77.8 | 92.0 | 70.00 | 84.09 | 83.4 |
| Deepseek-R1-671B[†] | 68.90 | 58.53 | 76.5 | 88.6 | 81.00 | - | - |
| GPT-4.1 | 66.29 | 59.68 | 78.4 | 87.8 | 64.00 | 85.54 | 77.8 |
| GPT-4o | 50.29 | 45.25 | 68.9 | 80.1 | 67.44 | 78.02 | 61.9 |
| o1-mini[†] | 64.20 | - | 71.3 | 80.8 | - | - | 87.9 |

Table 12: Full results on JETTS. Numbers in bold indicate that the judge reranking was helpful, i.e., performance is greater than baseline (greedy) performance.

| Benchmark | Generator Model | Baseline Performance | Oracle Performance | FARE-8B | FARE-20B | FARE-20B [CritiquePrompt] | gpt-oss-20B [CritiquePrompt] |
|---|---|---|---|---|---|---|---|
| MATH | Llama-3.1-8B-Instruct | 24.70 | 53.47 | **35.73** | **50.83** | **49.85** | **29.83** |
|  | Llama-3.1-70B-Instruct | 43.81 | 68.35 | **53.47** | **65.41** | **64.66** | **51.06** |
|  | Qwen2.5-32B-Instruct | 57.10 | 78.17 | **65.03** | **74.32** | **74.24** | **61.56** |
|  | Qwen2.5-72B-Instruct | 62.99 | 82.78 | **70.17** | **79.98** | **78.70** | **70.32** |
| GSM8K | Llama-3.1-8B-Instruct | 85.67 | 96.44 | **92.04** | **94.77** | **94.77** | **93.78** |
|  | Llama-3.1-70B-Instruct | 95.53 | 98.48 | **96.37** | **96.74** | **96.74** | **96.06** |
|  | Qwen2.5-32B-Instruct | 95.22 | 98.56 | **96.21** | **96.29** | **96.74** | **95.75** |
|  | Qwen2.5-72B-Instruct | 95.68 | 97.88 | **95.98** | 95.75 | **95.98** | 95.53 |
| CHAMP | Llama-3.1-8B-Instruct | 29.26 | 60.00 | **34.07** | **44.07** | **42.22** | **35.93** |
|  | Llama-3.1-70B-Instruct | 47.41 | 71.48 | **51.85** | **58.52** | **56.67** | **55.56** |
|  | Qwen2.5-32B-Instruct | 75.19 | 85.56 | 70.00 | **77.78** | **79.26** | 74.81 |
|  | Qwen2.5-72B-Instruct | 71.48 | 85.56 | 70.00 | **73.70** | **74.81** | 67.78 |
| MBPP | Llama-3.1-8B-Instruct | 54.50 | 76.46 | **59.79** | **68.78** | **68.25** | **63.49** |
|  | Llama-3.1-70B-Instruct | 65.08 | 83.07 | 62.17 | **67.72** | **68.78** | **67.99** |
|  | Qwen2.5-32B-Instruct | 75.40 | 84.13 | **76.72** | **80.42** | **79.37** | **79.10** |
|  | Qwen2.5-72B-Instruct | 76.19 | 84.66 | 75.40 | **78.31** | **78.31** | **78.04** |
| HumanEval | Llama-3.1-8B-Instruct | 63.35 | 79.88 | **64.02** | **74.39** | **74.39** | **68.29** |
|  | Llama-3.1-70B-Instruct | 75.61 | 90.85 | **76.83** | **88.42** | **85.98** | **84.76** |
|  | Qwen2.5-32B-Instruct | 81.10 | 93.29 | **83.54** | **91.46** | **90.24** | **87.80** |
|  | Qwen2.5-72B-Instruct | 82.32 | 93.90 | **86.59** | **90.24** | **90.85** | **86.59** |
| BCB | Llama-3.1-8B-Instruct | 31.67 | 56.84 | **34.82** | **41.84** | **41.23** | **39.30** |
|  | Llama-3.1-70B-Instruct | 45.44 | 62.63 | 43.86 | **46.93** | **47.54** | **45.88** |
|  | Qwen2.5-32B-Instruct | 45.53 | 65.18 | **47.02** | **49.39** | **48.95** | **48.25** |
|  | Qwen2.5-72B-Instruct | 46.67 | 60.18 | **47.54** | **49.04** | **49.30** | **48.25** |

### D.7 ADDITIONAL JETTS RESULTS.

In Table 12, we present results full results on JETTS. Concretely, we present (a) a prompt ablation, denoted `[CritiquePrompt]`, where we prompt FARE-20B and gpt-oss-20B for an critique and judgment. We additionally report results for different generators than Llama-3.1-8B-Instruct. Note that unlike FARE-20B, gpt-oss-20B does not natively support prompting without intermediate CoT, making an even comparison with our results presented in Fig. 3 unfeasible. Notably, FARE-20B is the only judge to improve performance over greedy across all generators and benchmarks, regardless of prompt. While FARE-8B is a relatively strong judge, it does not improve generator performance universally, struggling with larger generators on harder benchmarks. The trend of small evaluators struggling in helping larger generators was noted originally in JETTS. Across the board, FARE-20B improves in performance over gpt-oss-20B, sometimes by significant margins (e.g., 49.85 vs 29.93 for Llama-3.1-8B-Instruct MATH performance).

## E PROMPTS AND EXAMPLES

### E.1 OUR EVALUATION PROMPTS

Below we provide our evaluation prompts for pairwise, step-level, and verification evaluation, along with our direct judgment evaluation prompt for pairwise.

---

**Pairwise evaluation prompt for FARE**

```
### System Prompt

Please act as an impartial judge and evaluate the quality of the
responses provided by two AI assistants to the user prompt displayed
below.  You will be given assistant A's answer and assistant B's
answer.  Your job is to determine which assistant's answer is better.
If assistant A is better, output [A]. If assistant B is better,
output [B].

Here are some rules for evaluation

(1) When evaluating the assistants' answers, identify any mistakes
or inaccurate information.  Focus on the content each response and
select the response that is logically sound and error free.

(2) If both responses contain inaccurate information, select the
response that arrives at the correct response

(3) Avoid any biases, such as order of responses, length, or
stylistic elements like formatting

Before outputting your final judgment, provide an explanation of your
judgment.  Your explanation should discuss why your chosen response
is better based on the evaluation criteria.  The explanation should
concretely discuss strengths and weaknesses of both answers.

After outputting your explanation, provide your final judgment.  Use
the following format:

Explanation:  Your explanation here

Verdict:  Your final verdict

### User Prompt

[User Question]

{question}

[The Start of Assistant A's Answer]

{response_a}

[The End of Assistant A's Answer]

[The Start of Assistant B's Answer]

{response_b}
```

```
[The End of Assistant B's Answer]
```

**Direct judgment pairwise evaluation prompt for FARE**

```
### System Prompt

Please act as an impartial judge and evaluate the quality of the
responses provided by two AI assistants to the user prompt displayed
below.  You will be given assistant A's answer and assistant B's
answer.  Your job is to determine which assistant's answer is better.
If assistant A is better, output [A]. If assistant B is better,
output [B].

Here are some rules for evaluation

(1) When evaluating the assistants' answers, identify any mistakes
or inaccurate information.  Focus on the content each response and
select the response that is logically sound and error free.

(2) If both responses contain inaccurate information, select the
response that arrives at the correct response

(3) Avoid any biases, such as order of responses, length, or
stylistic elements like formatting

Output your final judgment directly.  Do not output any explanation
or rationale for your decision.  Use the following format:

Verdict:  Your final judgment

### User Prompt

[User Question]

{question}

[The Start of Assistant A's Answer]

{response_a}

[The End of Assistant A's Answer]

[The Start of Assistant B's Answer]

{response_b}

[The End of Assistant B's Answer]
```

**Step-level evaluation prompt for FARE**

```
### System Prompt

Please act as an impartial judge and evaluate the quality of the
response provided by an AI assistant to the user prompt displayed
below.  You will be given the assistant's solution to a math problem,
which is split into steps, starting with a <step [step number]> tag,
where [step number] is indexed from 0.  Your job is to identify which
step an error occurs, if an error is present.  When evaluating the
solution, consider each step separately.  Evaluate the content of
each step for correctness.  If you encounter a mistake at <step [step
number]>, output [step number] as your Verdict.  If the full response
is error free, then select step number -1.  Avoid any biases, such as
length of step, or stylistic elements like formatting.

Here are some rules for evaluation.

(1) The assistant's answer does not need to be complete or arrive
at a final solution.  You may receive a partially complete response.
Your job is to assess the quality of each step.
```

(2) When evaluating the assistant's answer, identify any mistakes or inaccurate information. Focus on the content each step and determine if the step is logically valid.

(3) For each step, you should provide an explanation of your assessment. If you find an error, describe the nature and cause of the error.

(4) Avoid any biases, such as answer length, or stylistic elements like formatting.

Before providing an your final verdict, think through the judging process and output your thoughts as an explanation After providing your explanation, you must output the corresponding step number with an error. Use the following format:

Explanation: Your explanation here

Verdict: The step number with the error or -1 if no error occurs

### User Prompt

[User Question]

{question}

[The Start of Assistant's Answer]

{response}

[The End of Assistant's Answer]

---

**Reference-based verification evaluation prompt for FARE**

### System Prompt

Please act as an impartial judge and evaluate if a response provided by an AI assistant (candidate answer) is consistent with a provided reference answer. Your job is to determine is the assistant's response is consistent with the reference answer.

If the response is consistent, output [A].

If the response is incorrect, output [B].

Here are some rules for evaluation.

(1) Refer to the given reference answer and determine if the candidate's answer is consistent with the reference answer.

(2) The reference answer is always correct and the question is perfectly valid. Take the reference answer as the ground truth.

(3) When determining if the candidate's answer is consistent with the reference answer, only compare the final answer. Ignore any potential errors in the reasoning processes.

(4) Some answers may be expressed in different ways, such as some answers may be a mathematical expression, some answers may be a textual description, as long as the meaning expressed is the same. Before making a judgment, please understand the question and the reference answer first, and then judge whether the candidate's answer is consistent with the reference answer.

(5) Some answers may consist of multiple items, such as multiple-choice questions, multiple-select questions, fill-in-the-blank questions, etc. Regardless of the question type, the final answer will be considered correct as long as it matches the standard answer, regardless of whether the reasoning process is correct. For multiple-select questions and multiple-blank fill-in-the-blank questions, all corresponding options or blanks must be answered correctly and match the standard answer exactly to be deemed correct.

```
Before outputting your final judgment, provide an explanation of
your judgment.  Your explanation should discuss why the response
is correct, incorrect, or invalid.  The explanation should
concretely discuss reasons for your judgment.  After outputting your
explanation, provide your final judgment.  Use the following format:

Explanation:  Your explanation here

Verdict:  Your final judgment of [A] or [B]

### User Prompt

<|User Prompt|>

{question}

<|The Start of Assistant's Answer|>

{response}

<|The End of Assistant's Answer|>

<|The Start of Reference Answer|>

{reference}

<|The End of Reference Answer|>
```

## E.2   SAMPLE EVALUATION RUBRIC

Here, we provide a sample rubric that was hand-written for SWE-Rank (Reddy et al., 2025). SWE-Rank data consists of contrastive pairs for training retrieval models. We re-purposed this data into a binary verification task, asking the evaluator if the retrieved code snippet was relevant for editing given a user request. "Positive" samples were assigned "Correct" labels, and "Negative" samples were assigned "Incorrect" labels.

### Example hand-written rubric for code retrieval samples

```
Here are some rules for evaluation

(1) Determine if the function provided by the assistant is a relevant
candidate for editing given the user's instruction

(2) A relevant function is one means that needs to be modified in
order to address the issue described in the user's instruction

(3) Modifying a relevant function does not mean is is sufficient
to resolve the user's issue.  That is, it is ok if modifying the
function does not completely resolve the user issue, but it should
make progress towards issue resolution.
```

