# OpenReview forum: "Foundational Automatic Evaluators: Scaling Multi-Task Generative Evaluator Training for Reasoning-Centric Domains"
_ICLR.cc/2026/Conference — ICLR 2026 Poster_

### Official Review · Reviewer_Z7Xv · 2025-10-16

**Soundness:** 3
**Presentation:** 4
**Contribution:** 4
**Rating:** 8
**Confidence:** 4

**Summary:**

This paper presents FARE (Foundational Automatic Reasoning Evaluators), a family of 8B and 20B parameter models trained to perform automatic evaluation across multiple tasks including pairwise comparison, step-level error detection, reference-based and reference-free verification, and single rating. The authors curate a 2.5M sample dataset spanning these evaluation tasks and multiple reasoning domains, then train their models using iterative rejection sampling supervised fine-tuning (RS-SFT). The paper evaluates FARE on seven static benchmarks and three practical downstream applications: inference-time reranking, verification during RL training, and domain-specific continual fine-tuning for code evaluation. The results demonstrate that FARE-8B challenges larger specialized evaluators, while FARE-20B establishes new state-of-the-art performance among open-source evaluators, even surpassing some 70B+ parameter models on certain tasks.

**Strengths:**

- The paper addresses a timely and important problem of building multi-task evaluators that can handle diverse evaluation scenarios, which is increasingly critical as LLMs become integrated into various applications.
- The data curation strategy is thorough and well-designed, combining 1.4M existing samples with 1.1M synthetic samples using both programmatic error injection and generate-then-grade approaches across multiple domains.
- The training methodology using iterative rejection sampling SFT is simple, stable, and scalable, avoiding the computational complexity and training instability of RL-based approaches while achieving competitive or superior results.
- The evaluation is comprehensive, covering seven challenging static benchmarks (JudgeBench, ReasoningJudgeBench, PPE, RM-Bench, When2Call, ProcessBench, VerifyBench) and three practical downstream applications that demonstrate real-world utility.
- FARE-20B achieves near-oracle reranking performance on MATH (Figure 3) and substantially improves RL training outcomes, with up to 14.1% relative gains over string matching verifiers (Figure 4), demonstrating strong practical impact.

**Weaknesses:**

- The cold-start initialization procedure for Qwen3-8B-Base using Qwen2.5-32B-Instruct data is not well-justified, and the authors acknowledge this produces a weaker baseline than the post-trained Qwen3-8B (Table 9), raising questions about whether better initialization could further improve results.
- The paper lacks detailed analysis of failure modes or systematic error analysis that would help understand when and why FARE models struggle, particularly on the benchmarks where performance lags behind GPT-5 or larger models.
- While the 2.5M sample dataset is impressively large, the paper provides limited information about potential data quality issues, redundancy across sources, or how the distribution of tasks and domains was optimized.
- The comparison with Self-Taught Evaluators (STE) could be more direct and comprehensive, as STE represents closely related work but is only evaluated on a subset of benchmarks, making it difficult to fully assess the relative merits of the approaches.
- Some experimental details are missing or relegated to appendices, such as the specific prompt templates used for each baseline model, hyperparameter selection procedures, and computational costs of training.
- The paper does not discuss potential negative societal impacts or limitations of deploying automatic evaluators at scale, such as perpetuating biases present in training data or the risks of evaluator exploitation.

**Questions:**

1. Why was the cold-start initialization from Qwen2.5-32B-Instruct necessary for Qwen3-8B-Base, and have you experimented with alternative initialization strategies that might preserve more of the base model's capabilities?
2. Can you provide more detailed error analysis showing specific failure modes of FARE models, particularly on the benchmarks where they underperform GPT-5 or larger baseline models?
3. How did you determine the optimal mixing ratios for different data sources in your 2.5M sample training set, and did you experiment with dynamic reweighting during training?
4/ What is the computational cost comparison between training FARE using RS-SFT versus training comparable evaluators using RL-based methods like RLVR?
5. For the generate-then-grade approach, how did you select the 12 generator models, and did you analyze whether certain generator characteristics (model family, size, reasoning vs non-reasoning) led to more valuable training data?

---

> ### Author Response · Authors · 2025-11-18
> **Response to Reviewer Z7Xv (Part 1)**
>
> We thank the reviewer for their constructive review! We’re happy they found our work “timely and important”, with our data curation approach “thorough” and our evaluations “comprehensive”. Below, we respond to questions and critiques point-by-point.
>
> > The cold-start initialization procedure for Qwen3-8B-Base using Qwen2.5-32B-Instruct data is not well-justified… the authors acknowledge this produces a weaker baseline than the post-trained Qwen3-8B…have you experimented with alternative initialization strategies that might preserve more of the base model's capabilities
>
> In initial attempts to continually finetune Qwen3, we found the model not amenable to continued training. As discussed in Appendix B, this tracked with public discussions, where the creators of Qwen3 themselves say that the post-trained variants of Qwen3 are hard to continually train; See footnote 4 on page 20. We hypothesize this results from the fact that Qwen3 is extensively post-trained to the point of “overtraining”.
>
> The recommendation from the Qwen developers (see same footnote above) was to start training from the base model. To do so, we cold-start with correct judgment traces sampled from Qwen2.5-32B and continue finetuning. While the cold-start model lags Qwen3-8B (post-trained), we observe that FARE-8B outperforms Qwen3-8B in evaluation across multiple benchmarks and domains. Starting from the cold-start model allows us to leverage the inherent capabilities of the base model while sharpening the task distribution performance towards evaluation.
>
> We consider continual training of evaluators from a base model interesting future work. We did not explore strategies extensively due to compute and timeline constraints. However, as we state in Appendix B.2, we hypothesize that a targeted finetuning stage prior to evaluation-specific training may improve performance across tasks. We leave this exploration as an interesting avenue of future work.
>
> > The paper lacks detailed analysis of failure modes or systematic error analysis that would help understand when and why FARE models struggle, particularly on the benchmarks where performance lags behind GPT-5 or larger models.
>
> This is a valuable point; We believe the fundamental difference in performance between GPT-5 and FARE models stems from differences in the **generation ability**, i.e., the ability to solve problems, of the two models. GPT-5 is a frontier generator, exhibiting state-of-the-art performance on many reasoning benchmarks, whereas gpt-oss-20b is relatively weaker as a problem solver. This delta in generation ability translates to verification, as better generators tend to be better evaluators, as shown in [1,2,3].
>
> Here, we present a small case study on PPE, which consists of 5 unique splits covering knowledge, natural language reasoning, instruction following, coding, and math. We report GPT-5 and gpt-oss-20B’s performance on benchmarks that measure these abilities: MMLU-Pro (knowledge), GPQA-diamond (reasoning), IFBench (instruction following), LiveCodeBench (code), and AIME (math). We also report the performance on the corresponding PPE split.
>
> |  |  | Knowledge | NL Reasoning | Instruction Following | Math | Code |
> |---|---|---|---|---|---|---|
> | GPT-5 | Generation Ability | 87.0 | 84.0 | 71.0 | 92.0 | 70.0 |
> |  | Evaluation Ability (PPE) | 90.7 | 81.3 | 97.9 | 97.5 | 67.7 |
> | FARE-20B | Generation Ability | 72.0 | 61.0 | 58.0 | 62.0 | 65.0 |
> |  | Evaluation Ability (PPE) | 80.3 | 68.5 | 69.1 | 94.6 | 59.3 |
>
> As seen above, evaluation performance tracks closely with generation performance, indicating that any gains in GPT-5 performance stem largely from the fact that it is generally a more capable model than gpt-oss-20B, the base model of FARE-20B. We take generation benchmark results from https://artificialanalysis.ai/.

---

> > ### Author Response · Authors · 2025-11-18
> > **Response to Reviewer Z7Xv (Part 2)**
> >
> > > While the 2.5M sample dataset is impressively large, the paper provides limited information about potential data quality issues, redundancy across sources, or how the distribution of tasks and domains was optimized… How did you determine the optimal mixing ratios for different data sources in your 2.5M sample training set, and did you experiment with dynamic reweighting during training?
> >
> > We spent great efforts in building our synthetic dataset generation pipeline to produce high quality training data. This includes rigorous answer matching using string matching and model-based verifiers to determine response correctness. In terms of redundancy, we sample diverse responses from various generators to ensure generalization to model responses; see further discussion in our response to the generate-then-grade question. Existing data was curated from existing sources deemed to be successful in training evaluator models in the past, e.g., [4], or newer sources for targeted domains or tasks.
> >
> > We do not experiment with dynamic reweighting or source mixing. Instead, we opt for simplicity: We ensure our starting point data is sufficiently high quality, using the approaches outlined above, then simply use all data available with fixed proportions. We consider data-centric evaluator work, i.e., finding the optimal subset or mixture from a large pool of evaluator data for training, an extremely interesting line of future work.
> >
> > > The comparison with Self-Taught Evaluators (STE) could be more direct and comprehensive, as STE represents closely related work but is only evaluated on a subset of benchmarks, making it difficult to fully assess the relative merits of the approaches.
> >
> > Thank you for the suggestions. We compare against STE in real-world response reranking scenarios via JETTS [5], and find that STE lags even FARE-8B on some benchmarks. However, we agree that additional pairwise benchmark comparison would be valuable in contextualizing the differences between FARE and STE. We report STE performance on JudgeBench, RJB, and RM-Bench below, using publicly reported results from [6,7]. We report pairwise only benchmarks because STE is trained only to perform pairwise evaluations.
> >
> > |  | JudgeBench | RJB | RM-Bench |
> > |---|---|---|---|
> > | STE-70B | 48.3 | 38.64 | 73.6 |
> > | FARE-8B | 55.71 | 51.05 | 79.2 |
> > | FARE-20B | 64.29 | 57.05 | 90.5 |
> >
> > Despite similarities in training approach, we find that STE-70B underperforms FARE-20B and even FARE-8B by large margins. This likely stems from limitations of their small-scale synthetic data generation, as discussed in Section 3.2. We have added these results to Table 1.
> >
> > > Some experimental details are missing or relegated to appendices
> >
> > Yes, unfortunately due to space limitations, we chose to highlight comprehensive experimental results in the main body. We included references to appendices where appropriate as a compromise; we are happy to edit based on suggestions!
> >
> > > The paper does not discuss potential negative societal impacts or limitations of deploying automatic evaluators at scale, such as perpetuating biases present in training data or the risks of evaluator exploitation.
> >
> > This is a valuable point. We have included an ethics statement in our updated manuscript on page 19.

---

> > > ### Author Response · Authors · 2025-11-18
> > > **Response to Reviewer Z7Xv (Part 3)**
> > >
> > > > What is the computational cost comparison between training FARE using RS-SFT versus training comparable evaluators using RL-based methods like RLVR?
> > >
> > > It is hard to precisely quantify computational cost differences between training FARE and other RL-based methods, as (1) multiple baselines only train on a subset of the tasks that FARE is trained on, (2) baselines use an unequal amount of training data. However, we present a case study compared against RL-trained StepWiser.
> > >
> > > StepWiser uses RL to train a step-level evaluator, requiring 5 days of training time: “We run the training for 800 steps, which takes approximately 5 days on 8 A100 GPUs” with a per-step prompt batch size of 1024. This means that it takes 5 days to train on ~800K unique samples.
> > >
> > > In contrast, we are able to train FARE-8B on 2.5M samples in 3.5 days on comparable hardware. That is, wall-clock time between the two methods is similar, but RS-SFT enables FARE-8B training on 3x more training data. This, in turn, leads to gains in downstream performance; FARE-8B (a generalist evaluator) outperforms the **task-specific** StepWiser by 1.6 abs. points, with even larger improvements in very hard splits (+5.6 abs. points).
> > >
> > > As an additional bonus, if stable and efficient RL approaches emerge, our curated training mix can be immediately used to train an evaluator with extended RL. We hope to explore the intersection of data scaling and RL for evaluators in future work.
> > >
> > > > For the generate-then-grade approach, how did you select the 12 generator models, and did you analyze whether certain generator characteristics (model family, size, reasoning vs non-reasoning) led to more valuable training data?
> > >
> > > To select generators, we selected models that are both relatively strong and weak to cover a wide array of response types. Because reasoning settings often include long CoT, we include reasoning models as a part of the generators. Beyond these factors, we focused on generators that are relatively well used. We did not experiment with finding an optimal subset of generators. However, recent work [8] has shown that if evaluators are trained on limited response distributions, they do not generalize well. As a result, we aimed for wide coverage to encourage better generalization.
> > >
> > >
> > >
> > > [1] https://arxiv.org/abs/2504.03846
> > >
> > > [2] https://arxiv.org/abs/2410.12784
> > >
> > > [3] https://arxiv.org/abs/2509.17995
> > >
> > > [4] https://arxiv.org/abs/2409.14664
> > >
> > > [5] https://arxiv.org/abs/2504.15253
> > >
> > > [6] https://arxiv.org/abs/2505.10320
> > >
> > > [7] https://arxiv.org/abs/2505.13346
> > >
> > > [8] https://arxiv.org/abs/2509.23542

---

> > > > ### Comment · Reviewer_Z7Xv · 2025-11-27
> > > > **Response to Authors**
> > > >
> > > > Thank you for your comprehensive response! After reading the rebuttal, I will maintain my positive score.

---

### Official Review · Reviewer_MQC2 · 2025-10-24

**Soundness:** 3
**Presentation:** 3
**Contribution:** 2
**Rating:** 6
**Confidence:** 5

**Summary:**

The paper introduces FARE (Foundational Automatic Reasoning Evaluators), a family of large-scale generative evaluators (8B & 20B parameters) designed for multi-task, multi-domain reasoning evaluation. It curates a 2.5M-sample dataset covering diverse evaluation types like pairwise, step-level, and verification tasks across reasoning-heavy domains such as math, code, and tool-use. FARE is trained using an iterative rejection sampling supervised fine-tuning (RS-SFT) method, enabling stable, scalable evaluator training without expensive RL or teacher models. Results show FARE-20B surpasses even larger RL-trained or specialized evaluators, achieving near-oracle reranking on MATH and significant gains in RL training and code evaluation. The work establishes FARE as a strong, open-source foundation for general-purpose evaluators in reasoning-centric LLM applications.

**Strengths:**

1. The 2.5M-sample multi-task dataset is one of the largest curated for evaluation, spanning reasoning, code, math, and tool-use, enabling strong generalization across domains.
2. The use of RS-SFT (rejection sampling SFT) achieves performance comparable to RL-based methods while being computationally more efficient and stable.
3. Evaluations on 7 benchmarks and 3 real-world tasks (e.g., MATH reranking, RL training, code evaluation) show strong improvements over state-of-the-art baselines.
4. The release of FARE models (8B, 20B) provides a valuable community resource for scalable evaluator research.

**Weaknesses:**

1. Novelty is rather low. The authors themselves mention that the method is simple and is a minor modification of methods like STE and RAFT. The paper emphasizes empirical scaling but provides little theoretical analysis of why RS-SFT works better for evaluators compared to RL or DPO approaches.
2. While reasoning-centric, the dataset and evaluations largely focus on math, code, and tool-use; broader language understanding or multimodal evaluations are underexplored.
3. Though large-scale, the paper lacks detailed quantitative analysis of noise, bias, or consistency across synthetic vs. real data sources.
4. Comparisons are strong against RL-trained evaluators but do not include recent alternatives like bandit-based scaling or entropy-minimization approaches to evaluation.

**Questions:**

1. "Existing data lays a solid foundation, with 1.4M samples already dwarfing data scales found in recent work." Please quantify this properly. What recent work? What is the number of samples in that work?
2. "Generate-then-grade" method. What model is used to grade the generated responses? What is the accuracy of the grader model?
3. What is task-wise and domain-wise distribution of existing vs synthetic datasets?
4. "After grouping responses by correctness, we create verification and pairwise." This sentence needs to be reworded -- does not make sense to me.
5. Line 237: "Because the automatic evaluation setting is inherently verifiable". You have automatic verifiers for all kinds of tasks? not all domains (chat, safety, reasoning) are objectively verifiable.
6. benchmarks: does it make sense to evaluate using PPE Correctness where golden responses are generated
using a variety of weaker models, e.g., Gemma-2-9B (which is smaller than the FARE-20B model you train)? Also, how are golden responses for these benchmarks curated: RM-Bench, When2Call, ProcessBench, VerifyBench?
7. Table 1: JudgeBench and RJB -- does not make sense to report GPT5 numbers and not GPT-4o numbers here. Aguably, GPT5 must be better than GPT4o. And since GPT4o was used to gather golden responses for these datasets, GPT4o must have 100% accuracy?
8. How does your method compare with high-performing TTC methods that explicitly address confidence calibration and adaptive compute while performing generation and evaluation together, such as: Efficient Test-Time Scaling via Self-Calibration (Huang et al., 2025) https://arxiv.org/pdf/2503.00031 COME: Test-Time Adaptation by Conservatively Minimizing Entropy (Zhang et al., 2025) https://openreview.net/forum?id=506BjJ1ziZ

---

> ### Author Response · Authors · 2025-11-18
> **Response to Reviewer MQC2 (Part 1)**
>
> We thank the reviewer for their extensive review! We are glad they believe our work will serve as a “valuable community resource for scalable evaluator research”, with our models improving over state-of-the-art baselines. Here, we respond to their questions and critiques.
>
> > Novelty is rather low. … The paper emphasizes empirical scaling but provides little theoretical analysis of why RS-SFT works better for evaluators compared to RL or DPO approaches.
>
> We want to reiterate that our goal in this work is to produce strong, usable checkpoints to drive future research in automatic evaluation. We take a data-centric approach, and focus on dataset scaling. The choice of RS-SFT was a decision made based on trade-offs: RS-SFT allows for more stable scaling, enabling faster development iterations. While theoretical insights are important, we believe that theoretical analysis comparing RS-SFT to RL/DPO to be out of scope for this work.
>
> > While reasoning-centric, the dataset and evaluations largely focus on math, code, and tool-use; broader language understanding or multimodal evaluations are underexplored.
>
> To clarify, our training data mix also includes evaluation data that focuses on safety, chat quality, and natural language reasoning as shown in Figure 2. We also evaluated on RM-Bench, which evaluates how evaluators understand subtle and stylistic factors in responses, a form of natural language understanding beyond surface level factors.
>
> We agree that multimodal evaluation is an important area, and are exploring ways to scale evaluator training data for other domains as future work.
>
> > What is task-wise and domain-wise distribution of existing vs synthetic datasets? … Though large-scale, the paper lacks detailed quantitative analysis of noise, bias, or consistency across synthetic vs. real data sources.
>
> Below, we provide per-task and per-domain breakdowns based on synthetic and existing data; We have edited our paper to include this table in Appendix B.
>
> | Task       | Pairwise          | Step-level | Verification | Rating   |       |        |
> |------------|-------------------|------------|--------------|----------|-------|--------|
> | Synthetic  | 45.2%             | -          | 54.8%        | -        |       |        |
> | Existing | 27.2%             | 36.0%      | 19.7%        | 17.1%    |       |        |
> |            |                   |            |              |          |       |        |
> | Domain     | General Reasoning | Math       | Code         | Tool-use | Chat  | Safety |
> | Synthetic  | 28.3%             | 52.1%      | -            | 19.6%    | -     | -      |
> | Existing | <1%               | 50.4%      | 14.7%        | 1%       | 25.7% | 7.2%   |
>
> This breakdown follows the discussion in Section 3.1: Our synthetic data augments our existing data sources by (1) introducing more verification data, a newly relevant task, (2) introduces more reasoning pairwise data samples, and (3) allows us to use new, difficult questions from RL-specific training sets.
>
> Quantifying noise or bias is a difficult endeavor within datasets. We took efforts to include high quality data. For existing data, we relied largely on sources deemed to be successful in training evaluator models in the past, e.g., [1], or newer sources for targeted domains or tasks. For synthetic datasets, we took great care in setting up our generate-then-grade pipeline, ensuring reliable ways to grade responses with both string matching and model-based checkers. In addition, because evaluators have been shown to prefer longer responses, a form of length bias [2], we took care in selecting positive and negative responses as well. In particular,  we ensured that the positive and negative responses were ones that minimized difference in response length to mitigate a spurious factor in evaluations.
>
> In aggregate, we believe that the performance of FARE on RM-Bench serves as a valuable proxy for the “cleanliness” of our curated data. RM-Bench uses pairs of responses that differ in subtle or stylistic ways, measuring how robust evaluators are to biases such as style or length bias. FARE models perform extremely strongly in this category, with FARE 20B achieving a ~10% relative improvement over the next-best specialized evaluator (J1-70B).
>
> In terms of consistency, we emphasize that our two data curation approaches (existing and synthetic) **complement** each other: Synthetic data leverages verifiable data to target underrepresented domains and tasks. In all, we believe that our dataset has great synergy, as shown by aggregate performance not only on static benchmarks, but on practical, real-world scenarios as measured by our downstream evaluations

---

> > ### Author Response · Authors · 2025-11-18
> > **Response to Reviewer MQC2 (Part 2)**
> >
> > > "Existing data lays a solid foundation, with 1.4M samples already dwarfing data scales found in recent work." Please quantify this properly. What recent work? What is the number of samples in that work?
> >
> > Thank you for your suggestion. We agree this statement can be more precise; For reference, J1 uses 22K samples [3], while RM-R1 uses 64K training samples [4]. CompassJudger uses the largest training set at 900K samples [5]. Using just existing data (1.4M samples) represents a 55% increase in size over CompassJudger’s training. We have edited this sentence to include concrete examples.
> >
> > > "Generate-then-grade" method. What model is used to grade the generated responses? What is the accuracy of the grader model?
> >
> > For generate-then-grade, we rely on benchmarks with objective ground-truth answers. For each sampled response, we use a first pass with string-matching and symbolic checkers, like Math-Verify, to determine if the answer is correct. For questions deemed incorrect, we use a *reference-based evaluation scheme*, prompting Qwen3-30B-A3B to determine if the model generated response matches the ground-truth. We performed ~100 randomized spot checks of Qwen3-30B-A3B during this grading process, and found no instances where we disagreed with the model’s verification.
> >
> > > "Because the automatic evaluation setting is inherently verifiable". You have automatic verifiers for all kinds of tasks? not all domains (chat, safety, reasoning) are objectively verifiable.
> >
> > To clarify, the intent of this statement is to highlight that **evaluators themselves can be trained easily with RLVR because evaluation model outputs have discrete, i.e., verifiable, answer spaces**. For example, one can train a pairwise evaluator by matching its output to one of two valid answers (“A” or “B”). This holds across all evaluation tasks considered in FARE, e.g., integers 1-5 for single rating or yes/no for verification.
> >
> > This is distinct from the process of actually obtaining these ground-truth answers in the first place. Here, domains like chat quality or safety are inherently open-ended, and obtaining an evaluation must be done without any “objectively correct” answer in mind.
> >
> > Our intention was to say that **if** one has evaluation training data, i.e., has obtained ground-truth evaluation answers, one can train an evaluator easily with RLVR. We have revised this statement in our new version for clarity, and thank the reviewer for their question.
> >
> > > benchmarks: does it make sense to evaluate using PPE Correctness where golden responses are generated using a variety of weaker models, e.g., Gemma-2-9B (which is smaller than the FARE-20B model you train)
> >
> > We believe that PPE evaluation is valuable despite the relatively weak generators. Certain splits, such as MBPP+ and IFEval, pose a challenge to even the strongest evaluators, as shown in Table 4. While generator strength may correlate with benchmark difficulty, it does not mean the benchmark is trivial: GPT-5 only achieves 87.0% on PPE, compared to 94.6% on solving competition-level problems, e.g. AIME. As such, PPE serves as a foundational benchmark for evaluating evaluators.
> >
> > A small note on model size: FARE-20B, as a mixture-of-experts model, only has 3.6B active parameters.
> >
> > > Also, how are golden responses for these benchmarks curated: RM-Bench, When2Call, ProcessBench, VerifyBench?
> >
> > RM-Bench was created via careful prompting and editing of responses using GPT-4o. When2Call responses are produced by Mixtral 8x22B. ProcessBench samples responses from 12 generators, such as Qwen2.5-Math-72B, Qwen2.5-72B-Instruct, and Llama-3.1-70B. VerifyBench samples from 22 generators, including DeepSeek-R1,V3, and Claude-3.7-Sonnet.

---

> > > ### Author Response · Authors · 2025-11-18
> > > **Response to Reviewer MQC2 (Part 3)**
> > >
> > > >Table 1: JudgeBench and RJB -- does not make sense to report GPT5 numbers and not GPT-4o numbers here… since GPT4o was used to gather golden responses for these datasets, GPT4o must have 100% accuracy?
> > >
> > > Because JudgeBench and RJB are pairwise benchmarks, each sample consists of one correct response and one incorrect response to a seed question, both generated by GPT-4o. Because each sample **must** contain an incorrect answer, the seed questions are inherently ones that are difficult for GPT-4o to answer, i.e., have a relatively low pass@1 score.
> > >
> > > The authors of JudgeBench [6] find that these samples are also difficult to judge: “if a model struggles to consistently generate correct, coherent responses to a challenging question, it will also struggle to distinguish between its correct and incorrect responses.”
> > >
> > > Indeed, this is borne out in evaluation, as reported in [6,7]: GPT-4o achieves 56.57 on JudgeBench and 47.18 on ReasoningJudgeBench, far lagging GPT-5 and FARE-20B. As such, we opt to report the stronger model, GPT-5, for comparison against the frontier.
> > >
> > > > How does your method compare with high-performing TTC methods that explicitly address confidence calibration and adaptive compute while performing generation and evaluation together…
> > >
> > > This is an interesting question. We believe our approach can complement calibration-based approaches. In our mind, the key distinction between the reviewer’s suggested works and our work is the degree of access to the generator that is required: The method in `Efficient Test-Time Scaling via Self-Calibration` requires access to the generator logits, whereas COME requires the ability to adapt the generator at test-time. In short, both methods require direct access to the generator model at inference-time.
> > >
> > > Generative evaluators, on the other hand, allow practitioners to view generators as black-boxes. This is an advantage when one does not have access to the generator model, as is increasingly common with API-based model inference with providers like OpenAI, Anthropic, Google, etc.
> > >
> > > However, this is not to say that calibration-based approaches are incompatible with generative evaluators. We consider ways of merging external evaluators and internal, calibration-based approaches and quantifying compute trade-offs and benefits an interesting line of future work. One concrete idea is to use an external evaluator to externally “check” a generator’s confident responses, as a sort of “outside perspective”.
> > >
> > > [1] https://arxiv.org/abs/2409.14664
> > >
> > > [2] https://arxiv.org/abs/2310.07641
> > >
> > > [3] https://arxiv.org/abs/2505.10320
> > >
> > > [4] https://arxiv.org/abs/2505.02387
> > >
> > > [5] https://arxiv.org/abs/2410.16256
> > >
> > > [6] https://arxiv.org/abs/2410.12784
> > >
> > > [7] https://arxiv.org/abs/2505.13346

---

### Official Review · Reviewer_TEm7 · 2025-11-01

**Soundness:** 3
**Presentation:** 3
**Contribution:** 3
**Rating:** 6
**Confidence:** 4

**Summary:**

This paper proposes a novel methodology to generate data and train an LLM-based evaluator and verifier. By combining different types of evaluation and verification tasks, they collect lots of training data featuring both synthetic and human preferences for downstream training. Authors stress that their result is one the examples of larger scale data scaling for training the evaluator/verifier models compared to more recent approaches that utilized relatively smaller sets for RLVR finetuning.
Authors rely on iterative SFT training and train a few models in the experiments. Their extensive results show that their model outperform known competing baselines on the benchmarks designed for evaluators. Moreover, their models show solid improvements when used for RL training with LLM as a judge/reward model, and outperform rule based rewards.

**Strengths:**

* Well executed recipe without any perplexed tuning or parameterization
* If the model will be released, then it will be especially useful
* Multi-tasking across different eval formats was shown to work

**Weaknesses:**

* W.r.t. the last strength point, i would be great to see the ablation of multi-tasking and how that affects the performance on specific eval tasks. From my understanding, there is no such experiment in the paper.
* When compared with competing evaluator models from literature, its a bit unclear what were other models initialization ckpts i.e. either its llama or qwen. Such difference may introduce not very fair comparison if we care about the added value of the proposed data and multitasking. Adding some extra notes about what are other model's pretrained architectures will help to make this more clear.

**Questions:**

* How was the model selection / early stopping performed? Is the same checkpoint used for all the evaluations in the paper?

---

> ### Author Response · Authors · 2025-11-18
> **Response to Reviewer TEm7**
>
> We thank the reviewer for their careful and constructive review. We’re glad they found our work “well executed” and “especially useful”. We are delighted to inform the reviewer that we have received the necessary institutional approvals, and will update the final, public version of the manuscript with appropriate links to Huggingface repositories.
>
> Here, we respond to the reviewer’s questions:
>
> > would be great to see the ablation of multi-tasking and how that affects the performance on specific eval tasks
>
> Early in our model development, we performed this ablation with Qwen-2.5-7B with pairwise and step-level evaluation data mixes. Note that the training datasets that were used were a subset of a preliminary version of our final curated dataset.
>
> Concretely, we trained 3 checkpoints: [pair] on pairwise data only, [step] on step-level data only, and [pair+step] on combined pairwise and step-level data. Below we report evaluation results. We evaluate [pair] on JudgeBench and ReasoningJudgeBench (RJB), [step] on ProcessBench, and [pair+step] on all three benchmarks.
>
> |  | JudgeBench | RJB | ProcessBench |
> |---|---|---|---|
> | [pair] | 53.71 | 48.78 | - |
> | [step] | - | - | 55.32 |
> | [pair+step] | 58.00 | 51.11 | 55.81 |
>
> As we see above, the multi-task checkpoint outperforms training individual experts independently, indicating that evaluation has shared fundamental task structure that lends benefits in multi-task training. Armed with this insight, we developed FARE with multi-task training.
>
> We agree with the reviewer that this would be an interesting insight to include in our paper, and have added a section in Appendix D.4 with this result.
>
> > When compared with competing evaluator models from literature, its a bit unclear what were other models initialization ckpts
>
> We wish to emphasize that our goal was to train useful checkpoints for the open-source community. However, we agree that explicitly mentioning base models of baselines would be good for clarity; We have updated the description of baselines in the appendix in our updated manuscript.
>
> > How was the model selection / early stopping performed? Is the same checkpoint used for all the evaluations in the paper?
>
> We keep an internal validation set which contains a mix of reasoning evaluation data and use validation performance for early stopping. We use the same checkpoint for all evaluations, i.e., we **do not** select the optimal checkpoint on a per benchmark basis

---

### Author Response · Authors · 2025-11-18
**Thank you to all reviewers!**

We thank the reviewers for their constructive feedback. In particular, we are grateful that reviewers found our work **“address[ing] a timely and important problem”** (Z7Xv), with the resulting FARE models **“especially useful”** (TEm7) and **“a valuable community resource”** (MQC2). In all, reviewers found our work **“well executed”** (TEm7), with our data curation **“thorough and well-designed”** (Z7Xv) and our evaluations on **“comprehensive”** (Z7Xv), with results on **“7 benchmarks and 3 real-world tasks…show[ing] strong improvements over state-of-the-art baselines”** (MQC2).

---

**We assure reviewers that we have received the necessary institutional permissions to release the FARE checkpoints; A link will attached to the final, public version of the manuscript to appropriate Huggingface repositories**

We have taken into account reviewer comments and have edited portions of our paper with additional details, with all changes marked in blue. We respond to reviewer comments individually below, and hope to have a productive discussion period!

---

### Author Response · Authors · 2025-12-04

Thank you to the AC for their hard work this review cycle. To help facilitate the review process, we are providing a summary of  our submission. Our submission received positive initial reviews of 8, 6, and 6, with all three reviewers providing thoughtful reviews and asking follow-up questions.

We have responded point-by-point, and have edited our manuscript in corresponding sections. Overall, we believe that our responses have  answered reviewer questions and addressed concerns.

---

Reviewer Z7Xv (Score 8) sought clarification primarily about modeling and data generation. We responded point-by-point, and the reviewer affirmed that they would maintain their score:
> Thank you for your comprehensive response! After reading the rebuttal, I will maintain my positive score.

Reviewer MQC2 (Score 6) sought clarification on certain baselines as they relate to benchmarks, and asked questions about specific statements in our work. We believe our response should clarify any questions and misunderstandings, and have edited our paper in response to specific reviewer suggestions. We did not receive a response from Reviewer MQC2.

Reviewer TEm7 (Score 6) had a primary concern about a missing ablation study and asked about checkpoint selection and for additional details about baseline models to be included in the paper. We presented an ablation result to the reviewer, and have updated our paper to include the ablation and make suggested edits. We did not receive a response from Reviewer TEm7.

---

### Meta-Review · Area_Chair_WCkC · 2026-01-10

**Summary:**

FARE addresses a timely topic in current test-time scaling and RLVR pipelines -- the lack of efficient yet sufficiently accurate verifier. By curating a large amount of data with different formats and styles, FARE is able to achieve state-of-the-art performance as an automatic evaluator. It has great implications on domains like scalable oversight, LLM-as-judge.

All reviewers recommend acceptance, and I also find this paper a great addition to ICLR.

**Reviewer Concerns:**

No significant concerns have been raised and the authors did a good job in addressing the reviewers' concerns.

**Reviewer Scores:**

All reviewers give positive ratings.

---

### Decision · Program_Chairs · 2026-01-26

Accept (Poster)